# Identification and characterisation of pathogenic and non-pathogenic *FGF14* repeat expansions

Repeat expansions in *FGF14* cause autosomal dominant late-onset cerebellar ataxia (SCA27B) with estimated pathogenic thresholds of 250 (incomplete penetrance) and 300 AAG repeats (full penetrance), but the sequence of pathogenic and non-pathogenic expansions remains unexplored. Here, we demonstrate that STRling and ExpansionHunter accurately detect *FGF14* expansions from short-read genome data using outlier approaches. By combining long-range PCR and nanopore sequencing in 169 patients with cerebellar ataxia and 802 controls, we compare *FGF14* expansion alleles, including interruptions and flanking regions. Uninterrupted AAG expansions are significantly enriched in patients with ataxia from a lower threshold (180–200 repeats) than previously reported based on expansion size alone. Conversely, AAGGAG hexameric expansions are equally frequent in patients and controls. Distinct 5′ flanking regions, interruptions and pre-repeat sequences correlate with repeat size. Furthermore, pure AAG (pathogenic) and AAGGAG (non-pathogenic) repeats form different secondary structures. Regardless of expansion size, SCA27B is a recognizable clinical entity characterized by frequent episodic ataxia and downbeat nystagmus, similar to the presentation observed in a family with a previously unreported nonsense variant (SCA27A). Overall, this study suggests that SCA27B is a major overlooked cause of adult-onset ataxia, accounting for 23–31% of unsolved patients. We strongly recommend re-evaluating pathogenic thresholds and integrating expansion sequencing into the molecular diagnostic process.

Spinocerebellar ataxias (SCAs) are a group of autosomal dominant, slowly progressive disorders characterized by impaired coordination and imbalance resulting in ataxia of stance and gait, limb ataxia, dysarthria and oculomotor signs[1]. Cognitive impairment, tremor, rigidity, bradykinesia, dystonia, spasticity, and polyneuropathy are frequently associated features. So far, at least 40 different SCA subtypes, classified according to their underlying locus/genetic cause, have been reported[2]. This list includes repeat expansions of CAG trinucleotides encoding polyglutamine (PolyQ) repeats (SCA1, 2, 3, 6, 7, 12, 17) as well as noncoding repeat expansions of penta- or hexanucleotides (SCA10,

12, 31, 36, 37). Furthermore, point variants have been described in at least 28 distinct genes[1,2]. Among those, rare point variants or microdeletions leading to the heterozygous loss-of-function of *FGF14* had previously been reported as SCA27 (renamed SCA27A)[3–5].

More recently, heterozygous noncoding GAA/CTT repeat expansions in *FGF14* have been identified as a frequent cause of late-onset cerebellar ataxia (SCA27B) in Canada, Australia, and Europe[6–8]. *FGF14* encodes the fibroblast growth factor 14, a gene expressing at least eight different isoforms, according to the Ensembl database (https://www.ensembl.org). The Matched Annotation from the NCBI and

✉ e-mail: christel.depienne@uk-essen.de

EMBL-EBI (MANE) isoform (NM_004115.4), also called transcript 1, encodes a 247-amino acid (27 kDa) protein that is considered as the canonical sequence in Uniprot (https://www.uniprot.org/). However, this transcript is poorly expressed in all tissues according to the Genotype-Tissue Expression (GTEx) database (https://www.gtexportal.org/). The most abundant transcript (transcript 2; NM_175929.3) is brain-specific, with the highest expression in the cerebellum, and encodes a 252-amino acid (28 kDa) protein. The two protein isoforms differ in their N-terminus as a result of distinct transcription start sites, leading to the inclusion of different first exons[9]. The N-terminus of isoform 1 contains a nuclear localization signal and is predicted to localize in the nucleus. In contrast, isoform 2 localizes at the axon initial segment of cerebellar Purkinje neurons[10], where it regulates the activity of voltage-gated sodium[9] and potassium[11] channels. The repeat expansion responsible for SCA27B occurs at a short tandem repeat highly polymorphic in humans located in intron 1 of isoform 2. The expansion is usually described as a GAA (equivalent to AAG) repeat expansion according to its genomic context, but as the gene is on the reverse strand, the expansion consists of CTT repeats in the gene context. The expansion is particularly frequent in Canada with an incidence up to 60% of families with late-onset ataxia in this population, due to a probable founder effect[6]. The comparison of expansion sizes in patient and control populations has suggested that expansions above 300 AAG repeats are pathogenic with full penetrance while expansions between 250 and 299 repeats are associated with incomplete penetrance[6,7]. The consequence of the expansion is a reduction of *FGF14* expression from the expanded allele, likely leading to heterozygous loss-of-function (haploinsufficiency) of isoform 2 in the cerebellum[6].

In this study, we identify *FGF14* repeat expansions as the most frequently missed genetic cause of cerebellar ataxia in two cohorts of patients, one with available genome data, and the other one with inconclusive diagnostic analyses. This finding led us to analyze *FGF14* alleles in a total of 1007 individuals, including 169 patients from 148 independent families with cerebellar ataxia of unknown cause, 32 patients with other neurological disorders, 4 healthy family members and 802 control subjects, using a combination of long-range PCR, fluorescent gene fragment analysis and targeted nanopore sequencing. Our results provide molecular, phenotypic and mechanistic insights into how pure AAG repeat expansions lead to adult-onset cerebellar ataxia while expansions of other motifs are non-pathogenic and interrupted expansions may be less penetrant.

## Results

### Identification of *FGF14* expansions from genome data

We used STRling to identify STR expansions in short-read genome data of 80 patients with neurological disorders (76 from Germany and 4 from Spain), including 48 patients with cerebellar ataxia from 39 independent families (Fig. 1, Methods). This analysis detected outlier values associated with significant $q$ values indicating possible AAG expansion in intron 1 of *FGF14* (chr13(hg38): 102,161,567-102,161,726) in 22 of the 48 patients with ataxia (19/39 families; 49%) but only three out of 32 individuals with other neurological disorders (3/31 families; 10%). In four families with ataxia, genome data were available for more than one affected member and outlier values were consistently detected for all affected relatives. In addition, STRling detected an expansion of an AAGGAG (hexamer) motif instead of AAG (triplet) motif at the same locus in one family with ataxia. Altogether, 26 patients had a possible *FGF14* expansion, consisting in AAG repeats in 25, and AAGGAG repeats in one (Supplementary Data 1).

We set up LR-PCR and RP-PCR assays to validate these results. The existence of at least one large *FGF14* AAG allele (PCR fragment size ≥ 700 bp corresponding to a triplet repeat number ≥180, see "methods") was confirmed for 18 of the 26 individuals, all with cerebellar ataxia. The eight remaining individuals (all three with another

neurological disorder and five with ataxia) had a larger allele below 670 bp (i.e., ≤ 170 repeats) and were therefore considered negative for *FGF14* expansion/SCA27B. Of note, we also confirmed that all patients that had no outlier values detected by STRling only had small *FGF14* alleles (Supplementary Fig. 1A).

### Screening of *FGF14* expansions in a second cohort

We then used the LR-PCR and RP-PCR assays to analyze 125 additional individuals (109 new index cases, 12 affected family members and 4 healthy relatives) with cerebellar ataxia (Fig. 1, Fig. 2, Fig. 3A, B and Supplementary Fig. 2). Thirty-three of the 109 index cases (30%) and six affected family members had at least one AAG allele ≥ 700 bp. Taken together with the expansions detected from genome data, 56 individuals from 46 families out of 148 independent families analyzed (46/148, 31%) had an expanded AAG allele ≥ 180 repeats estimated from fragment size (67 estimated from gel analysis). The AAG expansions segregated with the disorder in all families, including at least two affected individuals available for genetic analysis (Fig. 2; Supplementary Fig. 3A). Five patients from four families had an expansion of the AAGGAG motif (Supplementary Fig. 2B). The AAGGAG expansion segregated in two affected individuals from a Spanish family but was present in only the index case but not the affected mother of a German family (Supplementary Fig. 3B).

### Targeted Nanopore Sequencing of *FGF14* expanded alleles

Since LR-PCR and RP-PCR failed to provide the precise count of AAG repeats at the *FGF14* locus, we developed a nanopore sequencing assay of LR-PCR amplicons. In parallel, we analyzed and compared the distribution of *FGF14* alleles in a control cohort composed of 802 subjects. In total, we sequenced *FGF14* alleles in 67 patients, 64 control individuals and three unaffected relatives (Fig. 1; Fig. 3C, D; methods). We observed a strong correlation between allele sizes determined by fragment analysis or gel electrophoresis and the median number of repeats calculated from nanopore data (Fig. 3E). Combining fragment analysis and nanopore sequencing of LR-PCR amplicons provided a comprehensive view of allele sizes in both patients and controls.

Overall, 42 patients from 34 families (34/148; 23%) had at least one allele composed of pure AAG repeats exceeding the current established threshold for pathogenicity (250 repeats) and were considered as having SCA27B (Fig. 2). The median number of repeats calculated from nanopore reads ranged from 254 to 937 AAG repeats (Supplementary Data 2). Thirty-one patients had a repeat number above 300 repeats (Fig. 2A) whereas 11 patients had a repeat number between 250 and 299 (Fig. 2B). Nevertheless, we noted that the thresholds of 250 and 300 repeats appeared somewhat arbitrary.

Six patients from five independent families showed biallelic repeat expansions (Fig. 3B). One patient had two expansions above the pathogenic threshold (E24-0221; M100781;#9; Supplementary Fig. 4). In three families, the largest allele was between 250 and 300 repeats and the lowest allele below 250 repeats: 222/278 (E16-0360), 196/284 (E20-0778), 165/272 (E23-0439). The fifth family (E19-1058) included three affected siblings; two siblings had 204/311 and 196/319 repeats whereas their affected sister only had one large pathogenic allele (325 repeats).

One patient (E20-0501) repeatedly exhibited two large alleles, one with ≥ 300 repeats and another between 250 and 300 repeats, in addition to a small allele. We checked that this individual only has two copies of *FGF14* (Supplementary Data 3), suggesting somatic variability of the expanded allele (Fig. 3A, C). More generally, a high degree of somatic mosaicism was observed in all individuals with repeat expansions, as evidenced by the positive correlation between allele size and standard deviation (Fig. 3F).

### Distribution of *FGF14* alleles in patients and controls

The 802 control subjects showed an overall different distribution of *FGF14* alleles compared to patients with ataxia (Fig. 4A, B), especially

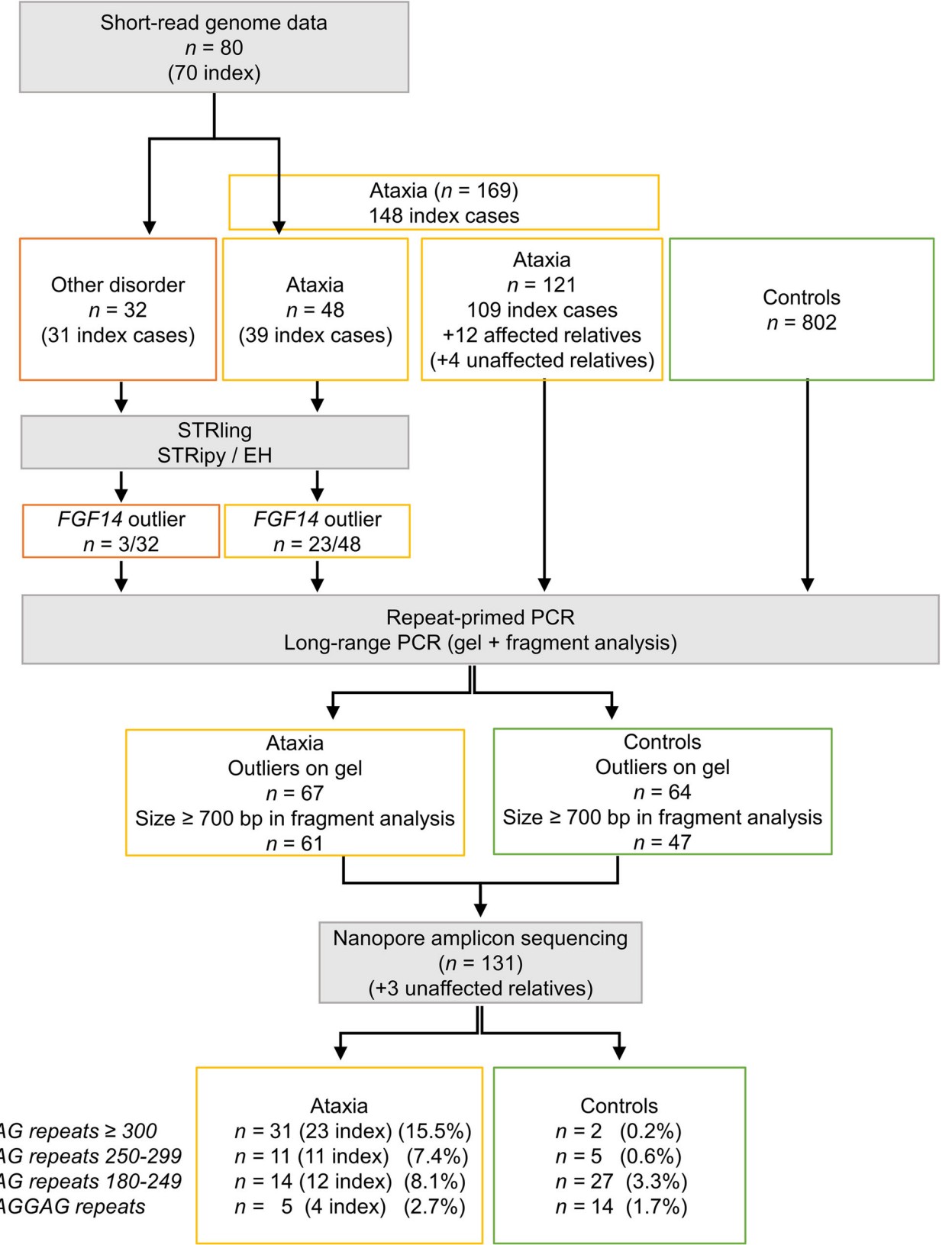

**Fig. 1 | Flowchart illustrating the design of the study.** This figure visually represents the sequential steps, methods used, and distribution of participants at each stage.

when considering only the larger allele (Supplementary Figs. 5A, B, Mann-Whitney, $p = 9.6e-6$). Large alleles composed of pure AAG ≥ 180 repeats were enriched in patients with ataxia, with alleles ≥ 250 repeats showing a more significant enrichment than intermediate alleles

(Fig. 4C, D; Supplementary Fig. 5C). Conversely, large AAGGAG expanded alleles were as frequent in patients as in controls (Fisher's test: $p = 0.74$, OR 1.16; Fig. 4A, B), confirming that these expansions are non-pathogenic, as already suggested[6,7,12,13]. Out of 21 control subjects

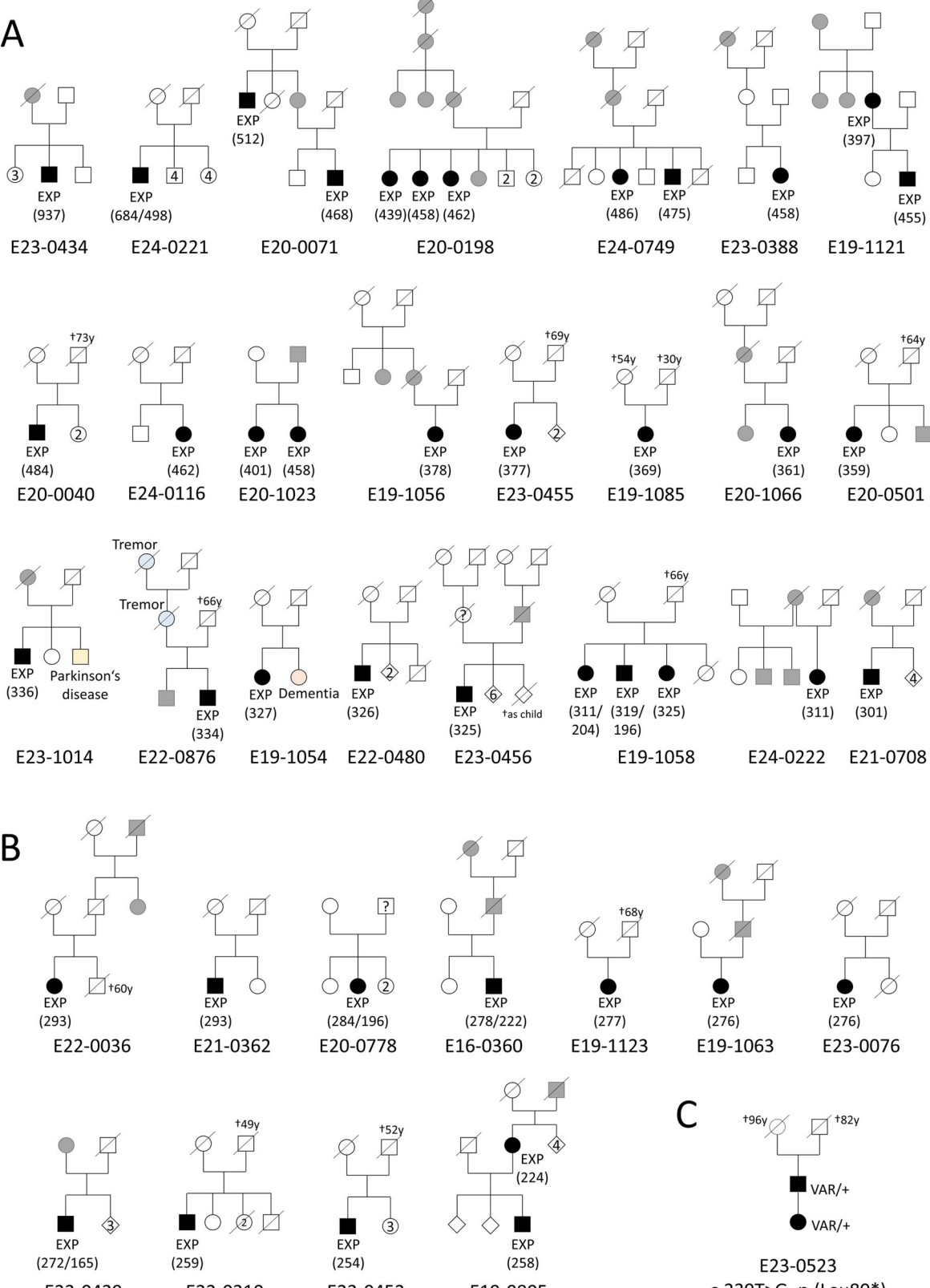

**Fig. 2 | Pedigrees of families with *FGF14* pathogenic repeat expansions.**
**A** Pedigrees of families in which at least one affected subject had a number of AAG repeats ≥ 300. **B** Pedigrees of families in which at least one affected subject had a repeat number comprised between 250 and 299. Black symbols indicate affected subjects examined and sampled in the study. Gray symbols indicate subjects

reported to be affected on history but that could not be examined. The number in brackets indicates the median number of repeats for the affected individuals of this family. The numbers in symbols indicate the number of siblings with the same sex. EXP indicates individuals with *FGF14* expansions. **C** Pedigree of the family with NM_175929.3: c.239 T > G; p.(Leu80*). VAR = variant i.e., c.239 T > G; *p*.(Leu80*).

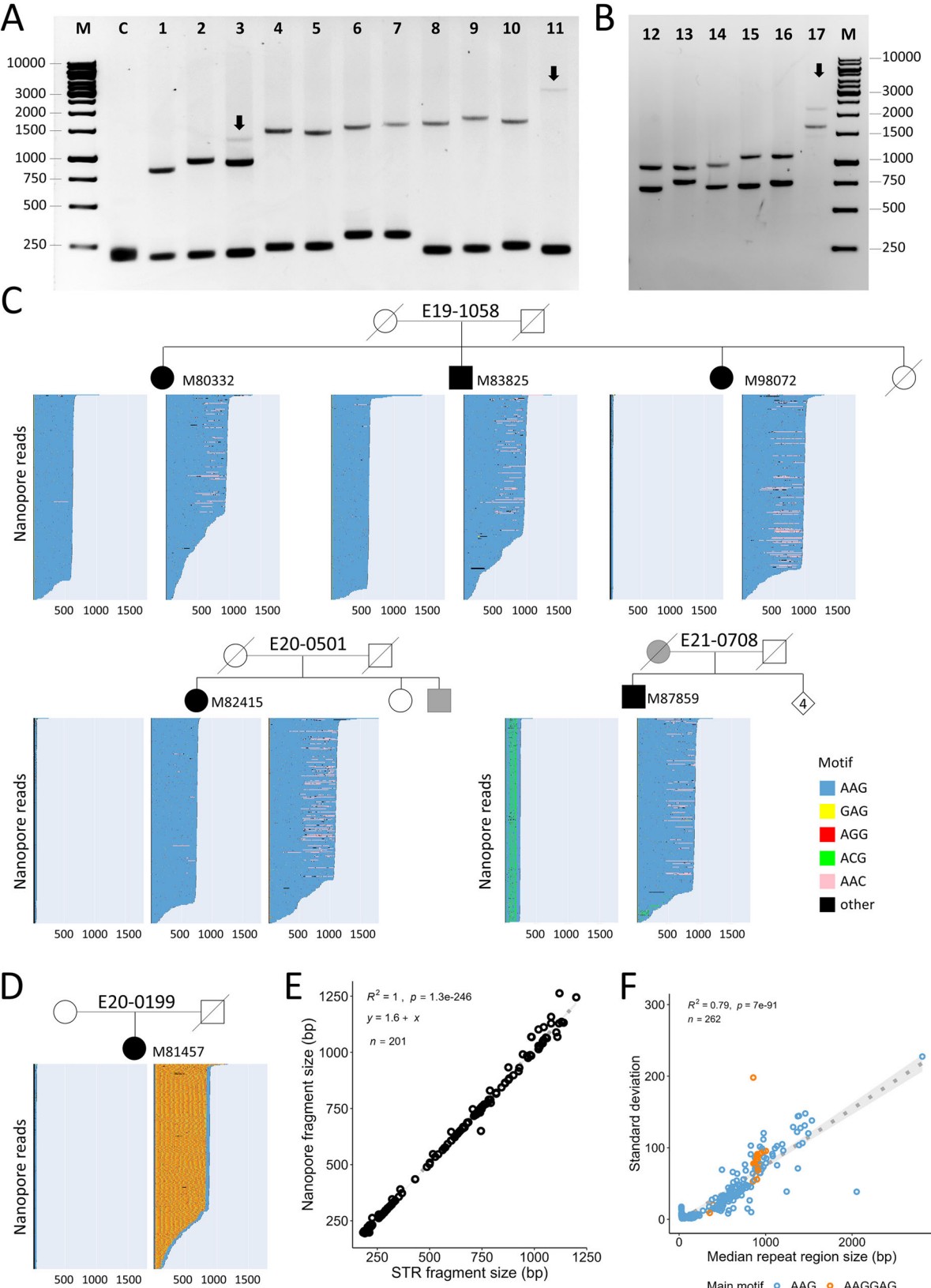

who had at least one allele above 250 repeats, 13 were composed of the non-pathogenic AAGGAG repeat motif (Supplementary Fig. 5, Supplementary Fig. 6). Eight control individuals had expansions ≥ 250 AAG repeats and only two, aged 46 and 70 years old at the time of sampling, had a pure AAG repeat expansion above 300 repeats (313 and 319 repeats respectively; Supplementary Data 2).

Comparing all alleles in patient and control sequenced samples, we observed true AAG interruptions (i.e., disrupting repeats in the middle; Supplementary Fig. 6) in smaller alleles only, with the size of these alleles being very stable, as shown by the limited standard deviation in repeat numbers (Supplementary Fig. 5D). Interruptions limited to 3' or 5' sides of the repeats have more limited impact on

**Fig. 3 | Analysis of *FGF14* repeat expansions in affected subjects using LR-PCR and nanopore sequencing. A** Gel electrophoresis of selected LR-PCR products spanning the *FGF14* (AAG) STR locus. From left to right, M: 1 kb ladder; C: negative control; lanes 1-2: expansions between 220 and 299 repeats (1-M79607, 2-M90982); lane 3: individual M82415 shows one small and two large alleles; lanes 4-11: expansions above 300 repeats (4-M87668, 5-M84267, 6-M93354, 7-M81456, 8-M95289, 9-M80996, 10-M80920, 11-M96642 (937 repeats)). **B** Gel electrophoresis of LR-PCR products showing biallelic expansions: lanes 12-M96652, 13-M97638, 14-M95716, 15-M83825, 16-M80332, 17-M100781. M: 1 kb ladder; LR-PCR assays and gel electrophoresis were performed at least twice with the same results. **C** Waterfall plots showing selected nanopore reads after separation of alleles based on their flanking regions (methods). 300 randomly chosen reads are displayed in each graph. Blue: AAG repeats; Yellow: GAG; Red: AGG; Green: ACG; Pink: AAC; Black: other. Panel above: segregation of *FGF14* alleles in family E19-1058 (three affected siblings, two with biallelic expansions and one with a single expansion of 325 repeats). Lower-left panel: individual M82415 (E20-0501) shows three different alleles: one small and two large (somatic mosaicism). Lower-right panel: individual M87859 (E21-0708) shows a pure AAG expansion (301 repeats) and a smaller allele with ACG interruptions. AAC streaks likely constitute errors of sequencing rather than true changes. **D** Nanopore reads from individual M81457 with an AAGGAG expansion. **E** Correlation between the median number of repeats detected by nanopore sequencing and expansion size estimated from fragment size analysis. **F** Standard deviation in allele size calculated from nanopore reads showing that somatic instability is positively correlated with expansion size. Blue: AAG; orange: AAGGAG main motif. For graphs (**E** and **F**), $R^2$ is the square value of the Pearson correlation coefficient (two-sided) and 95% confidence intervals appear in light gray.

repeat stability. They were equally frequent in both alleles (Fisher's test: $p = 0.40$) and not statistically different in patients and controls (Fisher's test: $p = 0.32$; Supplementary Data 4).

## Variability of the flanking region

Interestingly, the 5' region flanking the repeats (3' region in the context of the gene) was highly variable and drastically differed in expanded versus non-expanded alleles (Fig. 5, Supplementary Fig. 7). We observed seven different sequences following a constant CTTTCT motif (chr13:102,161,558-102,161,563) upstream of the repeats. These 5'-flanking sequences were either directly followed by AAG repeats, or preceded by short AG-rich sequences (e.g., AAGAAAGAG or AAGAG) that we considered as 'pre-repeat'. Pre-repeats were more frequently observed in larger (a2) alleles (Fisher's test: $p = 2e^{-8}$, OR 7.6, Fig. 5C, Supplementary Figs. 7C, D, Supplementary Data 4). The variable GTG sequence (present in the hg38 reference genome) was the most frequently associated with *FGF14* expansions although the range of repeats observed was highly variable (range: 36-684). GG and GGG sequences were only detected in expanded alleles (326-937 repeats). Conversely, the GTTAGTCATAGTACCCC was strikingly associated with small alleles (9-21 triplet repeats) only. Interestingly, one nearly identical sequence, differing only in the final two nucleotides (GTTAGTCATAGTACCAG), was associated with 203/207 repeats in both affected individuals of family E24-0752. This suggests that the four consecutive cytosines at the end of the sequence play a key role in preserving the stability of the adjacent repeats.

## Intermediate *FGF14* alleles

Fourteen patients from 13 families exhibited *FGF14* AAG repeat expansions ranging from 180 to 249 repeats (Fig. 2B, Supplementary Fig. 3A). According to the current threshold of 250 repeats for SCA27B diagnosis, these patients would be classified as negative. In family E19-0805, the affected mother (M79607) had a median repeat value of 224, prompting further investigation into alternative genetic causes for her ataxia. Despite analyzing genome data from both family members, no other ataxia-associated variants were identified, suggesting the *FGF14* expansion as the primary culprit. The comparison of alleles between 180 and 250 repeats in patients with ataxia versus control individuals, revealed an excess of expanded alleles in this range in patients (Fig. 4C, D). Segregation analysis was only possible for family E24-0752. We observed that the expanded alleles present in the two affected male siblings, consisting of 203 and 207 repeats and tagged by the GTTAGTCATAGTACCAG flanking region, retracted to 154, 164, and 164 repeats when transmitted to their unaffected offspring (Supplementary Fig. 4B). Among the patients with intermediate alleles, one female patient (M91143) had a variant of unknown significance in *PUM1* (NM_001020658.2: c.2180 T > C; p.(Ile727Thr)) alongside 236 AAG repeats in *FGF14* and one male patient (M87909) had a pathogenic SCA6 expansion (21 repeats) alongside 209 repeats in *FGF14*. This suggests the potential involvement of additional genetic or non-

genetic factors in the disease manifestation, with *FGF14* intermediate alleles possibly acting as susceptibility factors.

## Clinical comparisons

We divided patients with cerebellar ataxia for whom detailed clinical information was available into two main groups: 1) patients with at least one large allele ≥ 250 AAG repeats ($n = 42$; 'SCA27B-positive patients') and 2) patients with both alleles < 180 AAG repeats or composed of AAGAG repeats (group referred to as 'SCA27B-negative patients') ($n = 98$; Supplementary Data 5). Furthermore, we included the clinical data of a German family (affected father-daughter pair) with a previously unreported pathogenic nonsense variant in *FGF14* (SCA27A; NM_175929.3: c.239 T > G;p.(Leu80*); NM_004115.4(MANE): c.224 T > G; p.(Leu75*)) identified by routine exome sequencing.

Overall, most patients with *FGF14* expansion ≥ 250 AAG repeats (33/42; 78.5%) had a highly recognizable phenotype, characterized by the association of slowly progressive cerebellar signs accompanied by episodic symptoms of ataxia and/or downbeat nystagmus (DBN) that often present as first symptoms (Table 1, Supplementary Data 5). Nine patients exhibited cerebellar symptoms without episodic features or downbeat nystagmus (DBN).

When dividing SCA27B patients further according to the median repeat number, we observed that patients with ≥ 300 repeats ($n = 31$) and patients with 250–299 ($n = 11$) repeats had indistinguishable clinical characteristics (Table 1, Supplementary Data 4). For example, we observed a significantly higher occurrence of cerebellar oculomotor signs in SCA27B patients on first examination (95% in total; 94% for patients with ≥ 300 repeats and 100% for patients with 250–299 repeats, compared to 63% in SCA27B-negative patients). In particular, DBN was present in 50% (48% and 55%, respectively) at the first examination, compared to only 3% in patients negative for *FGF14*. Episodic symptoms were present in 93% of SCA27B patients (89%, and 100%, respectively) compared to 56% in the negative group. Patients with SCA27B had a lower occurrence of dysarthria on the first examination (21%; 19% and 27%, respectively) compared to 62% in the SCA27B-negative group. There was less cognitive impairment, as judged clinically, in patients with *FGF14* expansions (12%; 13%, and 9%, respectively in SCA27B patients compared to the 40% in the SCA27B-negative group at first examination). Patients with an intermediate allele (180-249; $n = 13$) exhibited greater clinical heterogeneity, with characteristics falling between those of SCA27B-positive and SCA27B-negative patients.

Among patients with ≥ 250 repeats, six patients exhibited a phenotype with additional signs or a more rapid disease course. Three out of these six patients had a biallelic expansion. Among patients with biallelic expansion, one female (M95716;#1 in Fig. 6A, B) had spasticity in the legs; another (M80332;#4) had cerebellar ataxia and an anxious/depressive disorder. She also had a stroke after first presentation from which she completely recovered (without significant change on SARA scores). SARA scores, however, may have been biased by anxiety-

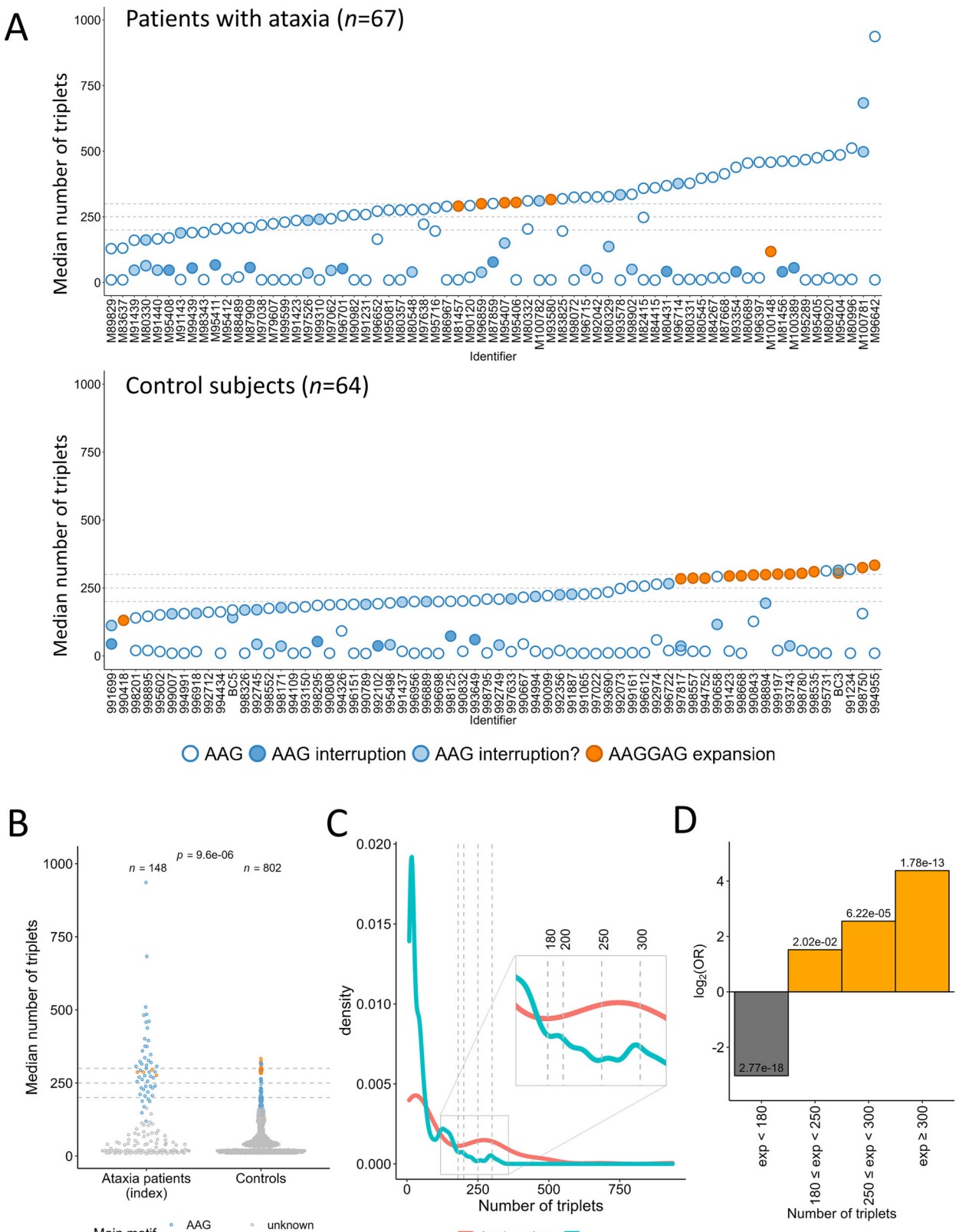

induced exacerbation of her balance problems. The third patient (M97638;#3) had a previous history of pulmonary sarcoidosis with muscular involvement (with no muscular involvement at initial examination and follow-ups). Patients with heterozygous expansions and atypical phenotypes included a female patient (M90120;#2) who had surgery for epidermoid tumor of the basal cisterns and seizures as well

as multiple meningiomas and developed leukoencephalopathy and dementia in later age; one female patient (M81456;#5) who had episodic choreatiform movements, increased muscle tone of the lower limbs, in addition to cerebellar signs and DBN; and one male patient (M80920;#6) with a slowly progressive cerebellar syndrome with mild autonomic symptoms over 20 years. Later on, he developed idiopathic

**Fig. 4 | Distribution of *FGF14* alleles in patients with cerebellar ataxia and control subjects. A** Median number of triplets of both alleles for the 59 patients with ataxia and 64 control individuals sequenced by nanopore sequencing. Pure AAG alleles are depicted as blue dots with a white fill. AAGGAG alleles appear in orange. Alleles with interruptions are depicted as blue dots with a dark blue fill. Alleles with interruptions limited to the 5' or 3' of the expansion are depicted as blue dots with a light blue fill. **B** Comparison of median allele sizes (larger alleles only; Mann-Whitney U test, two-sided) in index patients with cerebellar ataxia (*n* = 148) and control subjects (*n* = 802). Blue: AAG; orange: AAGGAG; gray: unknown main motif. **C** Density plot showing the different distributions of the number of triplets in the larger allele for index patients with cerebellar ataxia (*n* = 148; orange) and control subjects (*n* = 802; blue). **D** Log odds ratio according to repeat numbers of the larger allele (148 index patients with cerebellar ataxia and 802 control subjects) showing a significant enrichment of all classes of larger alleles ≥180 repeats in patients with cerebellar ataxia (Fisher's tests, two-sided, adjusted for multiple comparisons using Bonferroni correction; yellow: enrichment; gray: depletion). Each bar represents a single data point. Figures similar to (**B** and **D**) but considering all alleles appear in Supplementary Fig. 5.

Parkinson's disease and dementia. This patient previously had a clinical diagnosis of probable multiple system atrophy, cerebellar subtype (MSA-C)[14]. This would be the third patient with this diagnosis in whom a *FGF14* pathogenic expansion is identified[15,16]. However, in the context of our patient, a careful retrospective examination made the MSA-C diagnosis unlikely, and revealed that this patient actually had a typical SCA27B phenotype with later onset of second diagnoses that are frequent in elderly people. Detailed case reports of these individuals (clinical outliers; #1-6) as well as the three other patients with biallelic expansion (cases #7–9) are available in Supplementary Information.

The progression of SCA27B ataxia, assessed through SARA and ICARS scores, was generally slow, with a mean increase in SARA scores of 10.1 points over 30 years (Fig. 6A, B). Excluding the six patients considered as 'clinical outliers' resulted in even slower progression (6.7 points over 30 years; Supplementary Fig. 8AB). However, variability was high in both SCA27B groups (250 ≤ repeats < 300 and ≥300 repeats). The linear regression model suggests that disease duration accounts for only 6.5% of the observed variation ($R^2 = 0.065$), indicating that other factors exert a more substantial impact. Interestingly, a substantial proportion of patients reported worsening symptoms in the mornings (65%; 64% and 67% in SCA27B-positive groups compared to 33% in the SCA27B-negative group; Supplementary Fig. 8C). More than half of the patients who received 4-aminopyridine/fampridine reported symptom improvement (58%; 59% and 57%, respectively; Supplementary Fig. 8D). Treatment response to acetazolamide was also reported (38% and 60%, respectively; Supplementary Fig. 8E).

The patient harboring the pathogenic nonsense variant in *FGF14* (p.Leu80*; M96962) exhibited symptoms very similar to the groups of patients with ≥250 repeats. He had a slowly progressive cerebellar syndrome, episodic worsening of cerebellar symptoms and DBN. In addition, he also experienced episodic dystonia of the left hand. While he developed his first symptoms at the age of 36, his daughter had tremors at the age of 5 years and developed gait instability around 30 years.

### Correlations between repeat number and age at onset (AAO)

The mean AAO in our combined SCA27B-positive cohort (repeats ≥250) was 51.9 years (range 21–76; Table 1; Fig. 6) and 46.1 years (1–82 years) in the negative group. Patients with 250–299 repeats had a later AAO on average (57.2 years; range 34–76) than patients with ≥300 repeats (50.0 years; 21–75 years) although the difference was not significant (*p* = 0.13). Nine patients presented with an earlier form of the disease, with an onset before 40 years (21–38); all except one (M90982) had ≥300 repeats. Accordingly, we observed a significant inverse correlation between the number of AAG repeats and the AAO (Fig. 6F; Supplementary Fig. 8F). Nevertheless, 73% of the variation is independent of the number of AAG repeats ($R^2 = 0.27$). The variability of the AAO is illustrated by patient M90982 (258 repeats) who started to show symptoms at 34 years old whereas individual M91231 (259 repeats) experienced the first symptoms at age 72.

We conducted a meta-analysis to assess the correlation between AAO and expansion size, pooling data from our study and four prior studies[7,15–17]. We also included data of patients with truncating variants[4,5,18,19] or the F150S (F145S in MANE isoform 1) missense variant in *FGF14*[3,20] in the comparison (Table 1; Supplementary Data 6). Overall, we confirm a significant inverse correlation between AAO and expansion size (Fig. 6G; Supplementary Fig. 8G). The aggregated data show that patients with *FGF14* expansions between 250 and 299 repeats are significantly later affected on average (63.4 years; *n* = 36) than patients with ≥300 repeats (58.5 years; *n* = 115; *p* = 0.046; Fig. 6E). However, patients with expansions from both groups exhibited significantly later AAO compared to those with truncating (19.4 years, range: 2–47; *p* = 1.5e⁻⁴ and 2.5e⁻⁴) or F150S (20.5 years, range: 6–40; *p* = 1.5e⁻⁹ and 2.5e⁻¹²) variants (Fig. 6E; Supplementary Data 7). All statistical tests performed appear in Supplementary Data 4.

### Frequency of SCA27B in the German cohort

To assess the prevalence of SCA27B relative to other dominantly inherited ataxia subtypes, we compared the number of patients diagnosed in the ataxia outpatient clinic (Department of Neurology, University Hospital Essen). Among the diagnoses, SCA6 (*CACNA1A*) accounted for 40 patients, SCA27B for 36 patients, SCA3 (*ATXN3*) for 35 patients, SCA1 (*ATXN1*) for 17 patients, Episodic Ataxia 2 (point variant in *CACNA1A*) for 12 patients, SCA14 (*PRKCG*) for 7 patients, SCA2 (*ATXN2*) for 6 patients, and SCA8 (*ATXN8OS*) for 4 patients. Additionally, rarer forms such as SCA28 and SCA49 (two each), SCA7, SCA13, SCA15, and SCA27A (one each) were observed. This indicates that SCA27B ranks among the most prevalent SCA subtypes, representing approximately one-fifth of all diagnoses in patients with dominant cerebellar ataxia. The frequency of recessive ataxia subtypes was also lower than that of SCA27B. Excluding patients with variants of unknown significance, 17 patients had a biallelic expansion in *FXN* (Friedreich's ataxia), 8 a biallelic expansion in *RFC1* (Cerebellar Ataxia, Neuropathy, Vestibular Areflexia Syndrome, CANVAS), 7 had biallelic variants in *SACS* (ARSACS), 6 biallelic variants in *SYNE1* (SCAR8), 5 biallelic variants in *SETX* (AOA2) and 9 had rarer forms (4 patients with Tay Sachs disease, 3 patients with 4H-Syndrom and 2 patients with SCAR17). Compared to the total number of patients in our cohort (autosomal dominant and recessive), SCA27B accounted for 17%, Friedreich's ataxia for 8%, and *RFC1*-CANVAS for 4%. These findings independently support the high frequency of SCA27B diagnoses observed in another German cohort[21].

### Secondary structures associated with *FGF14* expansions

We used CD spectroscopy to assess the potential of secondary structure formation of the different *FGF14* antisense repeat expansions AAG and AAGGAG as well as the complementary sense sequences CTT and CTCCTT on DNA and RNA (Fig. 7A, B). The AAG-DNA 25-mer formed an antiparallel homoduplex while CD spectroscopy of the AAGGAG-DNA oligo revealed formation of a parallel homoduplex[22–25] (Fig. 7C). The RNA counterparts CUU and CUCCUU did not obtain any secondary structure under the tested conditions, confirmed by a single positive band at 270 nm[26] (Fig. 7C). Interestingly, the non-pathogenic AAGGAG-RNA oligo folded into a parallel guanine-quadruplex (G4) with a positive band around 260 nm, a negative band around 240 nm and positive values around 210 nm. The presence of a G4 was further confirmed by a G4-specific decrease in the stability detected, shifting from a parallel G4 structure (with 100 mM K) to a hairpin structure (with 100 mM Li)[27,28] (Fig. 7C). Of note, for the pathogenic AAG-RNA repeat we detected a CD spectrum related to an A-form RNA structure,

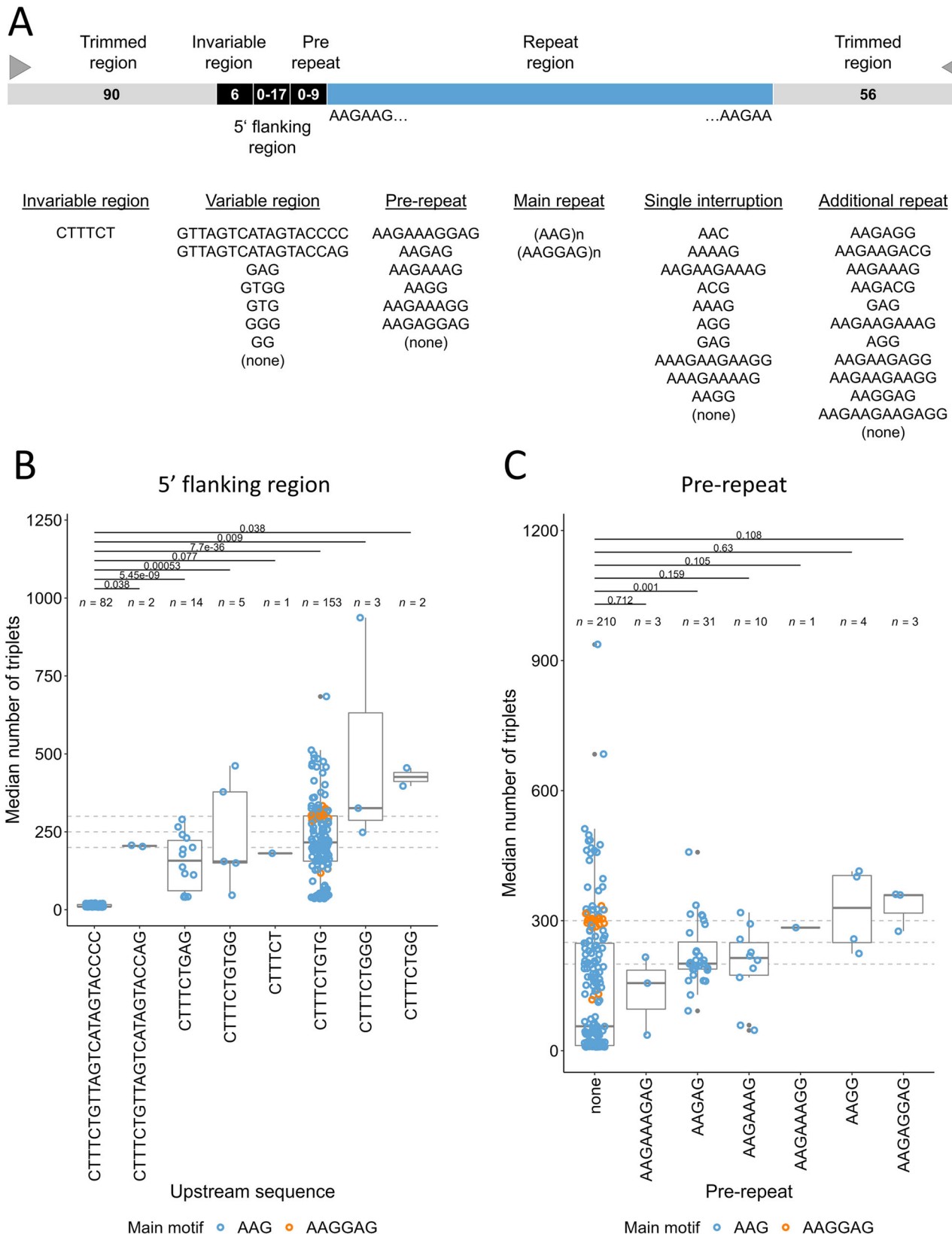

with a negative peak around 210 nm that reflects intra-strand interaction of RNA duplexes[29].

## Discussion

In this study, we identify intronic *FGF14* AAG expansions (SCA27B) as a major genetic contributor to cerebellar ataxia that had previously been

overlooked by short-read technologies. Bioinformatics tools such as STRling[30] can be used to detect this expansion from short-read genome data with a good predictive value (68%) and excellent sensitivity (100%; Supplementary Fig. 1). Although STRling largely underestimates the repeat count, the identification of a significant outlier value is an indicator of a possible expansion, requiring validation

**Fig. 5 | Effect of 5′ flanking regions on repeat instability. A** Schematic representation of the different parts composing *FGF14* repeat expansions. An invariable CTTTCT motif is usually followed by a variable 5′ region. A pre-repeat can be present in some individuals before the repeats. Some alleles are interrupted by one or several other motifs called interruptions. **B** Median number of triplets for each allele depending on the flanking region sequence. GTTAGTCATAGTACCCC is present in small alleles (≤21 repeats) only. Other sequences show higher number of repeats, suggesting higher instability of these associations. **C** Median number of triplets for each allele depending on the pre-repeat motif. Graphs displayed in (**B** and **C**) include both patients with ataxia and controls compared by Mann-Whitney U test, two-sided, followed by Holm correction for multiple testing. Graphs presenting data for patients with ataxia and controls separately appear in Supplementary Fig. 7. Box plot elements are defined as follows: center line: median; box limits: upper and lower quartiles; whiskers: 1.5× interquartile range; points: outliers. Blue: AAG; orange: AAGGAG main motif.

through an alternative method such as LR-PCR, RP-PCR and/or targeted long-read sequencing. A tool that can easily be implemented in routine diagnosis is ExpansionHunter[31] using the STRipy interface[32]. These targeted tools provide repeat number estimates higher than those obtained with STRling, but these values are largely below the pathogenic threshold and thus cannot be trusted. However, they are useful for analyzing the distribution of values within a population and identifying outliers (Supplementary Data 8), which can then be validated by more reliable methods.

The frequency of pathogenic expansions found in our combined ataxia cohorts, considering the previously established threshold of 250 repeats, was remarkably high (23%), equivalent to or even higher than previous reports[6,7,15–17,21,33–35]. The difference in incidence is possibly linked to the population studied but it could also result from the sensitivity of the genetic test used. We improved the LR-PCR conditions to minimize the amplification bias towards the smaller allele usually observed under standard conditions. This method could detect the largest expansion (937 repeats) reported so far. Current diagnostic strategies for SCA27B rely on sequential genetic tests based on the detection of normal and intermediate alleles (1000 bp being the usual limit of detection using fragment size analysis) combined with RP-PCR[36]. However, due to their inherent limitations, both tests may miss expansions that are unusually large and/or composed of other repeat motifs. A complete sequencing of *FGF14* alleles should thus become part of routine diagnostic procedures. Targeted nanopore sequencing is a cost-effective method that can easily be implemented to sequence *FGF14* expansions. Its known drawback is the higher error rate compared to other sequencing methods. However, errors can usually be easily distinguished from true sequence variations if the sequencing depth is sufficient, based on their random occurrence in single reads (errors) or presence in multiple reads (true variants). For example, nanopore sequencing leads to systematic errors (AAC instead of AAG) in long expanded reads. Moreover, smaller DNA fragments are preferentially sequenced by this technology. Since patients with *FGF14* repeat expansions show a high somatic variability that is correlated with expansion size, preferential sequencing of small fragments leads to underestimate repeat number in patients with large repeat expansions. To circumvent this problem, we used the median number of repeats instead of the mean, which is also closer to the size observed on gel.

The repeat expansion was mainly identified in patients with adult-onset cerebellar ataxia and was associated in 78.5% with a very recognizable phenotype that consisted of slowly progressive cerebellar ataxia with accompanying episodic symptoms of ataxia, often the initial manifestation, and/or downbeat nystagmus, as previously reported[37–40]. We further show that this characteristic SCA27B phenotype is consistent regardless of the number of repeats, with patients with as few as 191 repeats (M98343) exhibiting this phenotype. We also observed one family with a typical SCA27B phenotype in which the proband had 258 repeats and his affected mother 224 repeats. Genome sequencing did not reveal any other possible cause for the ataxia in this family. Furthermore, we also observed a significant enrichment of intermediate alleles (180-249 repeats) in patients with ataxia compared to the control population. This suggests that expansions below 250 may predispose to SCA27B, in line with two previous observations[21,38].

This finding contrasts, however, with that of a recent study that compared *FGF14* allele size in large patient and control cohorts without taking the sequence of the expansion into account[40]. The authors found no enrichment of pathogenic alleles in the 250–299 range, suggesting that molecular analysis strategies based on allele size only are likely to lead to misdiagnosis. This study also reported 10 patients who had *FGF14* alleles between 263 and 406 repeats and an additional possible diagnosis. We made a similar observation for a few patients with intermediate alleles, including one patient with a pathogenic SCA6 expansion identified after the inclusion in our study. Intermediate alleles and expansions above 300 repeats in some situations (e.g., when interrupted) could therefore contribute to disease only in combination with other factors, like it was recently shown for intermediate SCA17 alleles which lead to ataxia when associated with a heterozygous variants in *STUB1*[41,42]. Further studies involving larger patient and control cohorts are needed to refine the pathological thresholds, confirm the risk associated with intermediate alleles, and identify possible causes underlying the incomplete penetrance. Based on our observations, we recommend a reassessment of the pathogenic threshold in the range of 180–200 pure AAG repeats, which means that the frequency of SCA27B ataxia would be even higher (31%).

The mean AAO in our cohort was 51.9 years, but it strikingly varied from 21 to 76 years. Although most patients have a late-onset of first symptoms, nine patients presented with an earlier form of the disease, with an onset before age 40. As a result, SCA27B should not only be considered in late-onset ataxia as it may account for some cases of early-onset ataxia. We observed a positive correlation between the number of AAG repeats and AAO, but the contribution of repeat number to AAO is rather limited. Accordingly, we observed an important individual variation independent of the number of AAG repeats, as illustrated by the two patients with 258 and 259 repeats, but started showing symptoms at 34 and 72 years of age.

Consistent with the variability of the AAO, incomplete penetrance was observed in pedigrees of patients with 250–299 repeats, but also in those with ≥300 repeats. Incomplete penetrance above 300 repeats is further supported by the identification of pure repeat expansions above this threshold in two individuals of the HNR control population. Overall, the frequency of pathogenic *FGF14* expansions in the control population we studied, which is representative of the population from North Rhine-Westphalia, is 7 out of 802 (0.87%) when considering 250 repeats as a threshold, and up to 2.9% (23/802) taking 200 as cutoff, which is higher than the prevalence for rare diseases. Notably, two control individuals (46 and 70 years old) had pure AAG repeat expansion ≥300 (313 and 319 repeats, respectively). We did not have details about their neurological status and these individuals may develop symptoms of SCA27B later in life. Nevertheless, this suggests that incomplete penetrance exists within this range of repeats, and that the threshold for complete penetrance of pure expansions, if it exists, may thus be higher i.e., 320–335 repeats, as suggested by Rafehi et al.[7] Pellerin et al. demonstrated that *FGF14* repeat expansions tend to contract when inherited from the father, whereas they typically elongate when transmitted from the mother[6]. Accordingly, we observed a contraction of an allele of 203/207 repeats that was present in two affected male individuals of family E24-0752. Dynamic changes in repeat number upon transmission influenced by the sex of the parent

**Table 1 | Clinical features of patients with pure FGF14 expansions compared to a previously unreported FGF14 nonsense variant (SCA27A) and SCA27B-negative Patients**

| | FGF14 AAG expansion negative (n = 98) | FGF14 AAG expansion exp ≥ 250 (n = 42) | Adjusted p-value | FGF14 AAG expansion 180 ≤ exp <250 (n = 13) | FGF14 AAG expansion 250 ≤ exp <300 (n = 11) | FGF14 AAG expansion exp ≥ 300 (n = 31) | FGF14 truncating p.Leu80* (n = 2) |
|---|---|---|---|---|---|---|---|
| female sex - no. (%) | 49 (50) | 22 (52) | 0.8547 | 5 (38) | 5 (45) | 17 (55) | 1 (50) |
| age at onset, years (min-max) | 46.1 (1–82) | 51.9 (21–76) | 0.14 | 55.5 (9–77) | 57.2 (34–76) | 50.0 (21–75) | 20.5 (5–36) |
| age at last examination, years (min-max) | 60.6 (14–91) | 66.3 (27–85) | 0.0128 | 67.7 (49–85) | 68.0 (38–85) | 65.7 (27–85) | 60 |
| disease duration, years (min-max) | 14.6 (1–55) | 14.4 (2–34) | 0.475 | 12.3 (2–40) | 10.8 (3–30) | 15.7 (2–34) | 24 |
| **Symptoms at first examination** | | | | | | | |
| impaired gait (%) | 90/98 (92) | 37/42 (88) | 1 | 12/13 (92) | 9/11 (82) | 28/31 (90) | 1/1 |
| impaired stand (%) | 83/95 (87) | 22/42 (52) | **1.19E-04** | 9/13 (69) | 8/11 (73) | 14/31 (45) | 1/1 |
| cerebellar oculomotor signs (%) | 62/98 (63) | 40/42 (95) | **2.99E-04** | 8/13 (62) | 11/11 (100) | 29/31 (94) | 1/1 |
| - downbeat-nystagmus (%) | 3/98 (3) | 21/42 (50) | **7.91E-10** | 4/13 (31) | 6/11 (55) | 15/31 (48) | - |
| dysarthria (%) | 59/95 (62) | 9/42 (21) | **8.55E-05** | 6/13 (46) | 3/11 (27) | 6/31 (19) | - |
| cognitive impairment, according to examiner (%) | 30/75 (40) | 5/42 (12) | **8.85E-03** | 3/13 (23) | 1/11 (9) | 4/31 (13) | - |
| **Symptoms at last examination** | | | | | | | |
| impaired balance and gait (%) | 95/98 (97) | 42/42 (100) | 1 | 12/13 (92) | 11/11 (100) | 31/31 (100) | 1/1 |
| cerebellar oculomotor signs (%) | 85/98 (87) | 40/42 (95) | 1 | 12/13 (92) | 10/11 (91) | 30/31 (97) | 1/1 |
| - downbeat-nystagmus (%) | 6/98 (6) | 21/42 (50) | **6.53E-08** | 4/13 (31) | 5/11 (45) | 16/31 (52) | 1/1 |
| dysarthria (%) | 67/98 (68) | 16/41 (39) | **0.013** | 6/13 (46) | 6/11 (55) | 10/30 (33) | - |
| cognitive impairment (according to examiner) (%) | 59/83 (71) | 16/41 (39) | **0.005** | 7/12 (58) | 3/10 (30) | 13/31 (42) | 1/1 |
| pallhypesthesia (Rydel-Seiffer <6/8) (%) | 45/81 (56) | 16/35 (46) | 1 | 6/10 (60) | 6/10 (60) | 10/25 (40) | - |
| **Selected symptoms/features in medical history** | | | | | | | |
| episodic symptoms (any) (%) | 10/18 (56) | 25/27 (93) | **0.032** | 4/8 (50) | 8/8 (100) | 17/19 (89) | 1/1 |
| autonomic signs (any) (%) | 40/77 (52) | 11/37 (30) | 0.114 | 6/11 (55) | 4/10 (40) | 7/27 (26) | - |
| migraine (%) | - | 12/24 (50) | 1 | 2/9 (22) | 4/8 (50) | 8/16 (50) | - |
| seizures/epilepsy (%) | 6/24 (25) | 3/28 (11) | 1 | 6/24 (25) | 1/8 (12) | 2/20 (10) | - |

Clinical characteristics in bold are significantly different between groups. Comparisons were made using Fisher's tests, two-sided, adjusted for multiple comparisons using Bonferroni correction (see Supplementary Data 4 for details of all statistical tests performed).

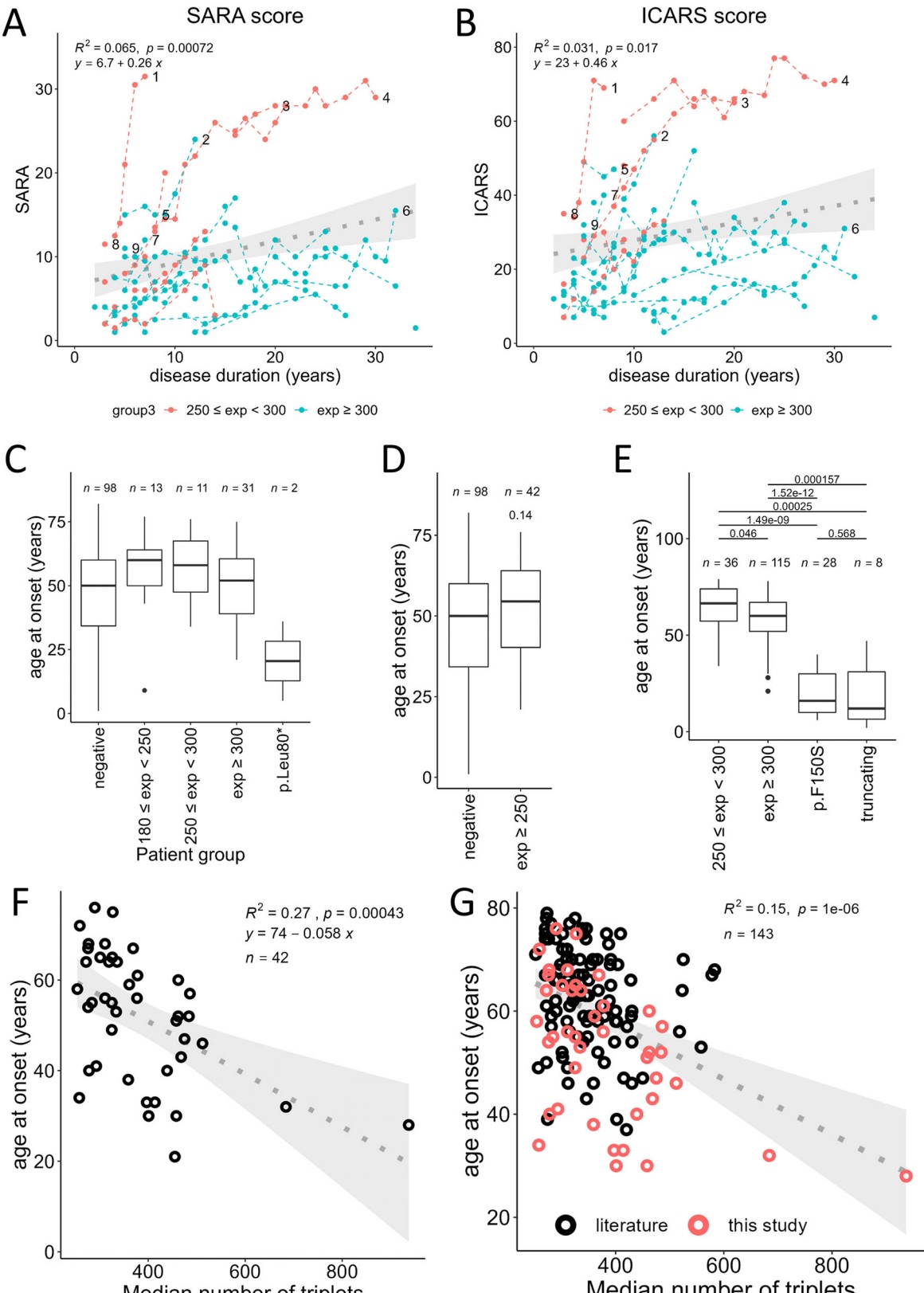

may underline the incomplete penetrance observed in some families and should be taken into account for genetic counseling.

Our study has revealed a more variable and complex structure of *FGF14* expanded alleles than previously foreseen, particularly obvious in control individuals. In particular, we suggest that the sequence of the expansion, including interruptions, pre-repeats and flanking

regions may influence the AAO and overall penetrance of the disease. For instance, individual M100781 (E24-0221), who has two large biallelic pathogenic AAG expansions (684/498 repeats) with the same 5' interruption composed of approximately 20 AAGGAG repeats (likely due to distant consanguinity), was the only affected of nine siblings. Although we could not study the segregation of these expansions in

**Fig. 6 | Disease progression and age at onset (AAO). A** SARA scores of 40 patients with *FGF14* expansions (*n* = 158 measurements). **B** ICARS scores of 40 patients with *FGF14* repeat expansions (*n* = 154 measurements). In graphs shown in (**A** and **B**) patients with 250-299 repeats and patients with ≥ 300 repeats appear in red and blue, respectively. Scores from the same patients at different time points are connected with dashed lines. Numbered last data points mark lines corresponding to atypical patients (#1–6) or patients with biallelic expansions (#1–3 and #7–9; Supplementary Information). SARA and ICARS scores are clinical rating scales used for semi-quantitative assessment of cerebellar ataxia (methods). **C** Comparison of the AAO in SCA27A/SCA27B-negative patients, patients with different expansion sizes: 180–249, 250–299 and ≥ 300 repeats; and two patients with p.(Leu80*). **D** Comparison of the AAO in SCA27A/SCA27B-negative patients, and SCA27B patients (repeat size ≥ 250 repeats). **E** Meta-analysis comparing the AAO in patients with expansions between 250 and 299 repeats, patients with ≥ 300 repeats, patients with nonsense or frameshift variants in *FGF14* or patients with p.Phe150Ser, showing that patients with pathogenic point variants (SCA27A) have an earlier age at onset than patients with repeat expansions (SCA27B). Box plot elements in **C**) to **E**) are defined as follows: center line: median; box limits: upper and lower quartiles; whiskers: 1.5× interquartile range; points: outliers; and comparisons were performed by applying Mann-Whitney U test, two-sided, followed by Holm correction for multiple testing. **F** Correlation between the AAO and AAG repeat number including only patients from this study. **G** Correlation between the AAO and AAG repeat number taking all patients from this study (red) and patients from previous studies (black) into account. For graphs **F**) and **G**), $R^2$ is the square value of the Pearson correlation coefficient (two-sided) and 95% confidence intervals appear in light gray.

unaffected relatives, both parents and several siblings are expected to be heterozygous carriers, which would mean that this particular heterozygous expansion is associated with reduced penetrance. The definition of interruptions (motifs, position in the expansion), the possible additive effect of other sequence variations (flanking regions, pre-repeats, additional repeats) within the expansion and their overall impact on SCA27B expression remain unclear and need to be further investigated in larger cohorts. The observation of six cases with biallelic expansions, three of whom have a more rapid and severe disease course, suggests an effect of the size of the smaller allele and/or other eQTLs at the locus. Finally, we anticipate that genetic variants linked to somatic instability of the repeats will also exist and modify the AAO. This possibly includes genetic variations at genes encoding DNA repair genes, like in other repeat expansion disorders[43].

The pathophysiological mechanism associated with *FGF14* expansions is likely a loss-of-function of isoform 2. This is supported by similar clinical features in patients with point mutations and repeat expansions. The p.Phe150Ser (F150S; F145S in isoform 1) variant, which was the first variant identified in *FGF14*[3], also leads to the loss of interaction with voltage-gated channels and a loss of FGF14 capacity to control their activity[44]. However, patients with nonsense or the F150S variant are on average earlier affected, suggesting that the loss-of-function of isoform 2 may be incomplete as a result of the expansion (i.e., expanded alleles would be hypomorphic compared to point mutations). Another possibility is that loss-of-function of isoform 2 would occur only when the repeats extend beyond an even higher threshold in neurons, as a result of aging. This hypothesis could explain the clinical variability observed in patients and also that intermediate alleles are only likely to expand beyond the pathogenic threshold in patients with poorer overall control of microsatellite stability. It is also possible that the number of repeats detected in peripheral tissues such as blood does not always reflect the number of repeats present in the brain or cerebellum.

The somatic variability of pure AAG repeat expansions is remarkable. Although positively correlated with expansion size, we also see an individual variability of this phenomenon. In particular, somatic variability is influenced by the presence of interruptions and by the sequence of the 5′ flanking region (Supplementary Fig. 5D; Fig. 5B). The association of specific flanking regions with repeat stability or instability has already been reported by Pellerin et al[45]. In this study, we extend this observation by showing that the four cytosines present in the flanking region associated with small alleles are crucial in this process. We hypothesize that these cytosines control the overall ability of the repeats to form secondary structures favoring repeat number amplification. This observation is to relate to secondary structures possibly formed by the repeats. We detected differences in the ability of pure pathogenic AAG and non-pathogenic AAGGAG repeats to form secondary structures at the DNA level but also at the RNA level. Noteworthy, the non-pathogenic AAGGAG RNA sequences can form a parallel G4 while pathogenic AAG repeats form an A-form RNA. G4 structures formed by other repeat expansions, including those found in *RFC1*/CANVAS, have been previously linked to a decrease in gene expression[46]. Furthermore, intronic AAG repeat expansions leading to Friedreich's ataxia were shown to form R-loops that impair the transcription of *FXN*[47–49]. However, AAG is the repeat motif present in *FXN* whereas in *FGF14*, the repeats present at the RNA level are CUU repeats (Fig. 7B). Hence, in the context of *FGF14*/SCA27B, there is a potential scenario where the capacity of AAGGAG repeats to generate G4 structures serves as a protective mechanism against the formation of other structures, such as R-loops on the complementary strand. A more comprehensive investigation of how these secondary structures affect gene expression is essential for *FGF14*, but requires studying this effect in the appropriate tissue or cell type, given the predominant expression of this gene in the brain and cerebellum.

In conclusion, this study reveals the sequences of pathogenic and non-pathogenic *FGF14* expansions. We suggest that pure AAG expansions are pathogenic from a lower threshold and account for 23–31% of patients with unsolved adult-onset cerebellar ataxia in European populations while interrupted expansions might be less penetrant, as suggested by the individual with biallelic interrupted expansions, and expansions composed of other hexanucleotide repeats motifs are non-pathogenic. Further studies looking for modifiers and diagnostic tests should therefore not only aim to assess the repeat number, but also include comprehensive sequencing of the expansion.

## Methods
### Patients & subjects
Patient inclusion was part of the project Identification of tandem repeat EXPAnsions in unsolved Neurological Disorders (EXPAND), which has received the approval of the ethics committee of University Hospital Essen (21-10155-BO). A written informed consent was obtained for all patients and subjects included in this study according to the Declaration of Helsinki. Patient/participant/samples were pseudonymized for the genetic study at each participating center. Participants receive no compensation for inclusion in the genetic study. Genetic and clinical data shared in the context of this study cannot be used to identify individuals. Researchers and clinicians from participating centers contributing either data or intellectual input were involved at all stages of the study from design, implementation, drafting, and revising the manuscript, and are coauthors of the article. At the time of the study, 76 patients with neurological disorders remaining without any identified genetic cause after an exome or a genome analysis were recruited from the Department of Neurology, University Hospital Essen, and had their genome sequenced. Among these, 44 patients from 35 independent families had spinocerebellar ataxia as a main clinical feature. Additionally, data of four families with cerebellar ataxia from Spain were included in the analysis. The *FGF14* repeat expansion was screened for in an independent cohort of 109 index cases with cerebellar ataxia without available genome data, 12 additional affected family members and four unaffected relatives. One patient had a point variant in *FGF14* (NM_175929.3: c.239 T > G; p.Leu80*) identified by routine exome sequencing. We collected self-reported information on the sex of the

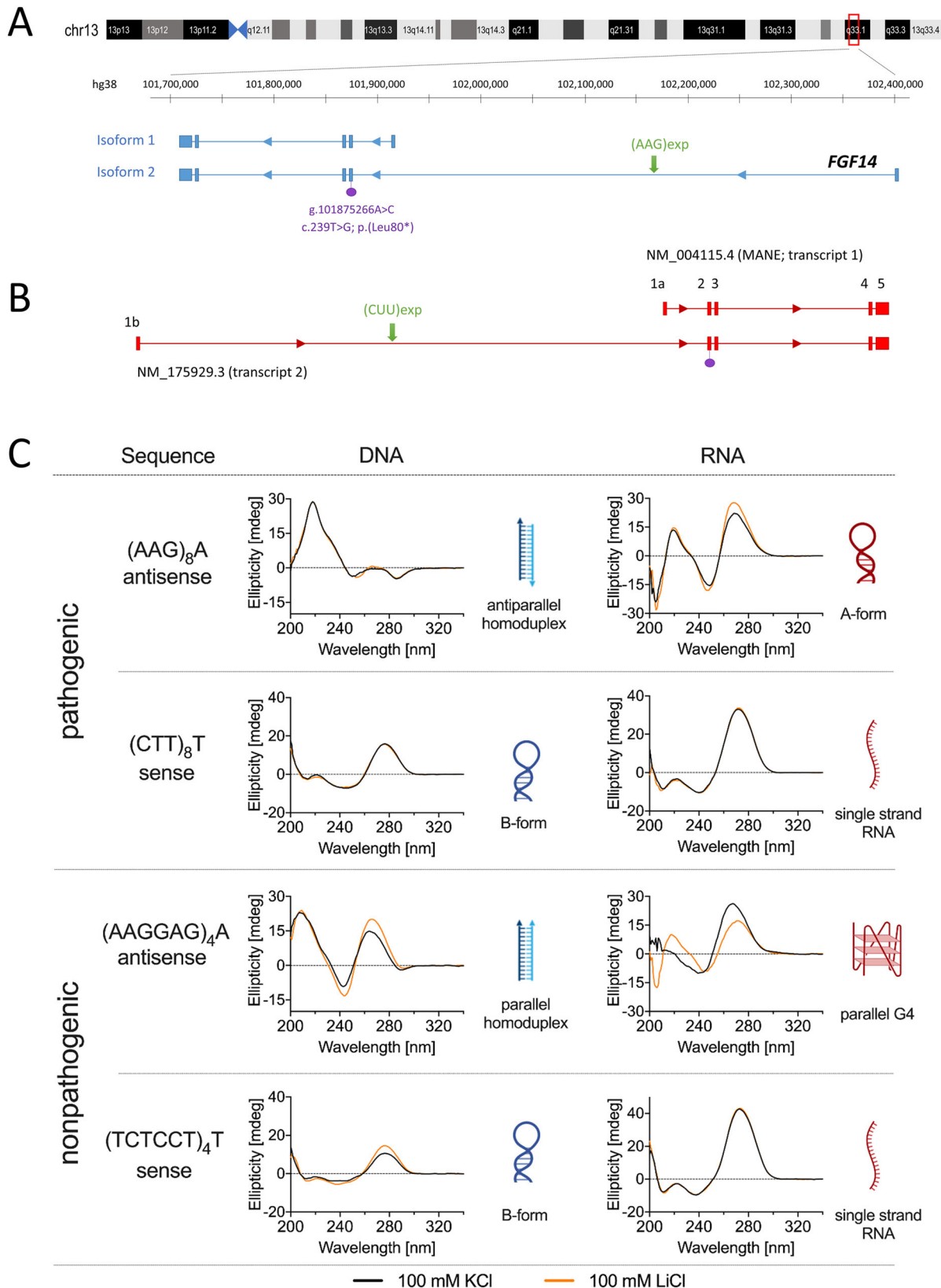

patients (from the patients' clinical file), but not on their gender. Although sex were not part of the inclusion criteria, the data on sex was used to evaluate the parental effect on repeat size during transmission to offspring. This study also included 802 control subjects: 30 anonymous blood donors and 772 participants from the Heinz-Nixdorf Recall (HNR) Study[50]. Sex and gender were not considered for the control subjects. No clinical information was available for the anonymous blood donors whereas individuals from the HNR study have been followed regularly for 20 years. However, neurological evaluation was not part of the standard follow up study and we therefore did not have access to the neurological status of individuals with pathogenic repeat expansions identified in this study.

**Fig. 7 | AAG and AAGGAG form different secondary structures at the DNA and RNA level. A** Schematic representation of the region on chromosome 13q33.1 containing the *FGF14* gene showing isoforms 1 (ENST00000376143.5; NM_004115.4) and 2 (ENST00000376131.9; NM_175929.3), which have alternative first exons. The gene is on the reverse strand. The green arrows show the location of the AAG expansion in intron 1 of isoform 2. The location of the previously unreported nonsense variant (NM_175929.3: c.239 T > G; p.Leu80*) reported in this study is indicated in purple. **B** Schematic representation of *FGF14* pre-mRNA isoforms 1 and 2. The expansion (green arrow) is composed of CUU repeats in RNA context. **C** Secondary structures formed by AAG and AAGGAG repeats at the DNA

and RNA level, assessed by circular dichroism spectroscopy. AAG repeats form an antiparallel homoduplex whereas AAGGAG repeats form a parallel homoduplex at the DNA level. At the RNA level, the AAGGAG repeats fold into a parallel guanine-quadruplex (G4) while AAG repeats adopt an A-form RNA structure. On the contrary, the CTT and TCTCCT repeats adopt a B-form and CUU and UCUCCU repeats did not form any particular secondary structure under the tested conditions. G-quadruplex and other DNA/RNA structures were created with BioRender.com released under a Creative Commons Attribution-NonCommercial-NoDerivs 4.0 International license.

## Clinical evaluation

Two assessors (L. M. and F. E.) conducted deep phenotyping through systematic reassessment of available medical records using a standardized datasheet. The age at onset of the disease was defined based on the medical history. The first symptom reported by the patient was used as age at onset, particularly, first episodic occurrence of gait ataxia, oscillopsia (as indication of downbeat nystagmus) or ataxia of gait. Impaired sense of vibration (reduced pallaesthesia) was defined as ≤ 5/8 on Rydel–Seiffer tuning fork at the medial malleolus (same threshold as used elsewhere[16]). Cognitive decline was assessed based on medical history and clinical judgment. It was documented as part of the Inventory of Non-Ataxia Signs (INAS)[51]. We used the Scale for the Assessment and Rating of Ataxia (SARA) and the International Cooperative Ataxia Rating Scale (ICARS), which are clinical rating scales used for semiquantitative assessment of cerebellar ataxia validated in multi-center trials, to monitor disease severity and progression. SARA[52] consists of eight items (gait, stance, sitting, speech, finger-chase test, nose-finger test, fast alternating movements, heel-shin test) with a score ranging from 0 (no ataxia) to 40 (most severe ataxia). ICARS[53] consists of 19 items and 4 subscales (postural and gait disturbances, limb ataxia, dysarthria, oculomotor disorders) with a score rating from 0 (no ataxia) to 100 (most severe ataxia). In instances where information was unavailable, NA was recorded in the standardized datasheet. Only patients with available information were taken into account for the calculations of percentages and statistical comparisons.

For clinical comparisons, data were stratified according to their genetic status: 1) SCA27B patients with AAG repeats ≥ 300; 2) patients with 250–299 repeats; 3) patients with an intermediate allele (180–249 pure repeats); 4) patients negative for SCA27B (i.e., less than 180 repeats). SCA27B-negative individuals had an exome analysis (or a panel analysis including all genes known in ataxia) prior to their inclusion in this study. Among these, one family (affected father-daughter pair) had however a previously unreported pathogenic nonsense variant in *FGF14* (SCA27A;NM_175929.3:c.239 T > G; p.(Leu80*); NM_004115.4(MANE):c.224 T > G; p.(Leu75*)) identified by routine exome sequencing. Clinical features of this family were included in the comparison study. No variant in *FGF14* was reported in the remaining SCA27B-negative patients.

## Genome sequencing & expansion calling

Short-read genome sequencing was performed at the Cologne Genome Center (Cologne, Germany) for 51 patients and as part of routine diagnosis at the Institute for Medical Genetics and Applied Genomics (University of Tübingen, Germany) for 25 patients. Additionally, data of four families with cerebellar ataxia from Spain were included in later analysis. Libraries were prepared with the DNA tagmentation based library preparation kit without PCR (Illumina DNA PCR-Free Prep Tagmentation; Illumina; reference 20041794), with 300–500 ng genomic DNA input. Library preparation was followed by clean up and/or size selection using SPRI beads (Beckman Colter Genomics). After library quantification (Qubit, Life Technologies) equimolar amounts of library were pooled. The library pools were quantified using the Peqlab KAPA Library Quantification Kit (Material Number: 07960204001) and

the Applied Biosystems 7900HT Sequence Detection System and then sequenced on an Illumina NovaSeq6000 sequencing instrument with a paired-end 2x150bp protocol. Fastq data were mapped to the hg38 reference genome using an in-house pipeline: raw sequencing data underwent preprocessing using cutadapt[54] to remove adapter sequences. Read mapping used bwa-mem[55], bwa-mem 2[56], and bwa-meme[57]. Duplicate reads were removed with samblaster[58]. Sorted and indexed CRAM files were generated by samtools[59]. In an initial phase, we compared the ability of ExpansionHunter DeNovo[60] and STRling[30] to detect known repeat expansions from short-read genome data including TTTTA/TTTCA repeat expansions in *MARCHF6*[61] and *STARD7*[62], and a full ( > 200) CGG repeat expansion in *FMR1* (Fragile X). Both tools performed similarly but we chose STRling based on its ability to detect a more accurate number of repeats compared to ExpansionHunter DeNovo. We then used STRling (version 0.5.2) to call short tandem repeats on the processed and mapped sequencing data at the genome-wide level. AAG is the motif detected by the bioinformatic tools (equivalent to GAA, avoiding redundancy of repeated motifs). In this study, we decided to keep this nomenclature, especially as the repeat expansion generally starts with an AAG motif after the pre-repeat (see Fig. 5A). In addition, known repeat expansions were called in parallel by ExpansionHunter[31] v5.0 (with and without off-target regions) and STRipy[32] (https://stripy.org/) using the extended mode. The *FGF14* reference region was defined as follows: chr13:102161576-102161726. We used off-target regions provided by catalog creator from STRipy (https://stripy.org/expansionhunter-catalog-creator). Variant catalog json files are available in https://doi.org/10.5281/zenodo.12542853.

## Positive predictive value and sensitivity

We calculated the positive predictive value (PPV) and sensitivity of the tools used for detecting expansions from short-read data: ExpansionHunter (with and without off-targets), STRipy (extended) and STRling. The formulas used were: $PPV = TP/(TP + FP)$ and $Sensitivity = TP/(TP + FN)$, where TP are the true positives, FP are the false positives and FN are the false negatives. We pre-calculated the 85th, 90th, 95th and 99th quantiles for the distribution of repeat sizes detected in 498 genomes (including the 80 reported in this study) by STRipy and both modes of ExpansionHunter. For each sample with repeats exceeding the quantile value and for each repeat number threshold (200, 250 and 300), we classified samples into TP, FN and FP, based on the comparison of the chosen threshold with the number of repeats quantified by different methods (nanopore sequencing or fragment analysis). STRling reports outliers detected based on significant qvalues, thus, classification of each sample into TP, FN and FP was achieved by assessing if the number of AAG repeats quantified by different methods (nanopore sequencing or fragment analysis) correlated outlier detection for each chosen threshold.

## Quantitative PCR (qPCR)

We used the qPCR mode of primer3Plus (https://www.primer3plus.com/) to design primers for quantitative amplification of exon 1a (isoform 1) and exon 1b (isoform 2) of *FGF14* (FGF14_ex1_iso2_F2-TGTCTAAGCTGCTGGATTGC and FGF14_ex1_iso2_R2-GCATATGCGTT

CCTTTGCTG; FGF14_Ex1_iso1F-ATCGCTAGCGGCTTGATCC and FGF 14_Ex1_iso1R-GAAGATGCGCACTTTGGAGAAG). qPCR experiments were performed in triplicates from 25 and 50 ng of genomic DNA using the KAPA SYBR® FAST Master Mix (Merck; KK4611). ASSRO2 on chromosome 15q11.2 (F-AGCGAAGCTCAGACATCATTTG, R-CAAACCTT-TAACAGCAGCTGACCTA) was used as the control region. qPCR reactions were run on a Lightcycler 480 (Roche) with the following thermocycling conditions: 95 °C for 10 min (1 cycle); 95 °C for 15 s and 60 °C for 1 min (45 cycles); and 37 °C for 30 s (1 cycle). Relative abundance was calculated using the formula $r = 2^{-\Delta\Delta Ct}$ with $\Delta\Delta Ct = (CT_{FGF14\_Ex1} - CT_{ASSRO2})_{individual\ tested}$ / mean $(CT_{FGF14\_Ex1} - CT_{ASSRO2})_{control\ individuals}$.

## Long-range PCR amplification

Repeat expansions at the *FGF14* locus were amplified by Long-Range PCR (LR-PCR) from genomic DNA extracted from blood using a protocol adapted from Rafehi et al[7]. Notably, we reduced the number of cycles of the LR-PCR to limit an enhanced amplification of smaller alleles. The amplification of *FGF14* repeat alleles was performed from 50 ng genomic DNA in 25 µl using the HotStarTaq DNA Polymerase (Qiagen; 203445) and 0.20 µM of each of the following primers: FGF14_RPP_ F1: AGCAATCGTCAGTCAGTGTAAGC; FGF14_LRP_ R1: CAGTTCCTGCCCACATAGAGC. The PCR program comprised an initial step at 95 °C of 15 min; followed by 28 cycles, each consisting of 30 seconds at 95 °C, 30 seconds at 60 °C and 2 minutes at 72 °C; and a final step at 72 °C of 10 min. LR-PCR amplicons were analyzed on a 1.3% agarose gel. The PCR was performed with a FAM-marked forward primer for gene fragment analysis on ABI 3130xl DNA Analyzer (Applied Bio systems). The size of *FGF14* alleles below 700–1200 bp were quantified using the GeneMarker software (SoftGenetics).

## Targeted sequencing with Oxford Nanopore Technologies (ONT)

LR-PCR products were sequenced for 134 individuals: 108 subjects (61 patients with cerebellar ataxia and 47 control subjects) with an allele ≥ 700 bp, 23 subjects (17 controls and 6 patients) with an allele between 487 and 685 bp (due to discrepancies between outliers detected on agarose gel and by fragment analysis) and 3 healthy relatives (family E24-0752). LR-PCR amplification was performed without FAM-labeled primer in a total volume of 75 µl. LR-PCR amplicons were purified using the DNA Clean & Concentrator-5 (Zymo Research; D4004). We use the SQK-LSK109 ligation-based sequencing kit (Oxford Nanopore) and the native barcoding protocol (EXP-NBD196) to prepare the libraries and multiplex samples, respectively. In brief, this procedure involves the following stages: 1) End-prep, where 200 fmol of each purified amplicon undergoes incubation with NEBNext Ultra II End Repair/dA-tailing Module Reagents (New England Biolabs; E7546L) at 20 °C for 5 min and 65 °C for 5 min in a 96-well plate; 2) Native barcoding ligation, comprising incubation with native barcodes and NEB Blunt/TA Ligase Master Mix (New England Biolabs; M0367L) at 20 °C for 20 min and 65 °C for 10 min; 3) Pooling of bar-coded amplicons and purification using AMPure XP beads (Beckman Colter; A63881); 4) Adapter ligation, involving a 10-minute incubation with Adapter Mix II Expansion/NEBNext Quick Ligation Reaction Module (New England Biolabs; E6056S) followed by clean-up with AMPure XP beads. All steps were executed in accordance with the manufacturer's recommendations. Approximately 15 ng of the final prepared library was loaded onto a MinION Mk1B R9.4.1 FLO-MIN106 flow cell, and nanopore sequencing was conducted for up to 24 hours, and monitored using the MinKNOW software.

Basecalling and analysis of nanopore data were performed using command line Snakemake[63] workflows available at GitHub (https://doi.org/10.5281/zenodo.12655177[64] and https://doi.org/10.5281/zenodo.12654084[65]). This workflow takes fast5 files as input and utilizes several tools to generate sequence_summary.txt, final_summary.txt, and

fastq.gz files. Guppy[66] (version 6.4.6) was used for basecalling, pycoQC[67] (version 2.5.2) and NanoPlot[68] (version 1.41.6) for quality control. The command line version of Guppy, guppy_basecaller, used the parameters: `--recursive --compress_fastq --do_r-ead_splitting --calib_detect --records_per_fastq 0 --enable_trim_barcodes`. After comparing the basecalling hac (dna_r9.4.1_450bps_hac.cfg) and sup (dna_r9.4.1_450bps_sup.cfg) models, the sup model was selected for further analysis. The pass-reads of samples sequenced on multiple runs were pooled. The initial fastq files were quality trimmed and filtered with BBMap[69] bbduk.sh using the parameters: `-Xmx2g qin=33 minlen=200 qtrim=lr trimq=10 maq=10 maxlen=100000`. Two 25 bp flanking sequences upstream chr13(hg38):102161532-102161557, ATATCAATATTCTCTAT GCAACCAA) and downstream (chr13:102161726-102161751, TAGAAA TGTGTTTAAGAATTCCTCA) of the repeat expansion were used to filter for all reads that had both flanking sequences, allowing 2 mismatches per flanking sequence using bbduk.sh with --literal and --edist parameters. Reads where both flanking sequences showed a -strand orientation were subsequently converted to their reverse-complement (+ strand) sequence. All other reads were discarded. In the next step, flanking sequences were trimmed off from both sides using Cutadapt[54] to leave only the repeat containing region, as well as the invariable and variable region (5′ flanking region) and pre-repeat. We considered only reads containing two AAG repeats (i.e., the sequence AAGAAG). Other reads were discarded. The workflow calculates statistics and generates plots (custom Python and R scripts) to characterize the nature of the repeat expansion in length and motif composition. Specific alleles (length and motif) can be defined manually for enhanced visualization.

For each sample, reads were visually inspected in Geneious Prime® 2019 (Biomatters Ltd.) for identification of the sequences corresponding to the invariable region, 5′ flanking region, pre-repeat, main motif, additional repeat motif, and interruptions. Differences between alleles (inclusion or exclusion of sequences, minimum or maximum length) were used to separate reads from different alleles into a1 (smaller), a2 (larger) and a3 (intermediate, mosaic cases). Plots were generated for the separated alleles using up to 300 random reads per allele. For the multiple reads obtained for each allele of a sample, we calculated the median size of the repeat region (after subtracting from the length of each sequence, the sizes of the invariable region, 5′ flanking region and pre-repeat), the median number of triplets and its standard deviation. Even for hexameric repeats (AAGGAG), we report the number of triplets to allow direct size comparison with AAG repeats. For comparisons with the sizes obtained by fragment analysis, we increased the median size by 146 bp, which were previously removed by trimming. Reads with interruptions were separated in two categories: true AAG interruptions (i.e., disrupting repeats in the middle) and interruptions limited to 3′ or 5′ sides of the repeats ("interruptions?"). Examples of interruptions can be seen in Supplementary Fig. 6.

## Repeat-primed PCR

*FGF14* AAG repeat expansions were amplified by repeat-primed PCR (RP-PCR) with 6-FAM labeled FGF14_RPP_F1 (AGCAATCGTCAGTCAG TGTAAGC), and non-labeled RPP_M13R (CAGGAAACAGCTATGACC) and FGF14_RPP_AAG_RE_R1 (CAGGAAACAGCTATGACCCTTCTTCTT-CTTCTTCTTCTT). AAGGAG expanded alleles were amplified using FAM-FGF14_RPP_F1, P3 (TACGCATCCCAGTTTGAGACG), and P3_AAG-GAG (TACGCATCCCAGTTTGAGACGAAGGAGAAGGA-GAAGGAGAAG). PCR was performed from 100 ng genomic DNA, with 0.8 µM primer FGF14_RPP_ F1, 0.8 µM primer RPP_M13R or P3, and 0.26 µM primer FGF14_RPP_AAG_RE_R1 or P3_AAGGAG using the HotStarTaq Master Mix (Qiagen; 203445). The PCR program consisted in 95 °C for 15 min, followed by 35 cycles (94 °C for 30 s, 61 °C for 30 s, and 72 °C from for 2 min) and a final extension step at 72 °C for 10 min. RP-PCR products were detected on an ABI 3130xl DNA Analyzer and analyzed using GeneMarker software (SoftGenetics).

## Statistics

Fisher's tests (two-sided) were performed to determine associations between: (i) a sample belonging to a class of triplet numbers (either in all alleles or just in the larger allele of an individual) and being an ataxia patient; (ii) a patient belonging to a class of SCA27B-negative/pathogenic expansion size or point mutations and presenting certain symptoms; (iii) a patient belonging to a class of *FGF14* pathogenic expansion size and having a certain family history presentation; (iv) a sample having an AAGGAG expanded larger allele and being an ataxia index patient; (v) a sequenced allele containing a pre-repeat and being a larger allele; (vi) a sequenced allele containing a 3' or 5' interruption and being a small allele; (vii) a sequenced allele containing a 3' or 5' interruption and being from a control. Odds ratios were $\log_2$ transformed and indicate enrichment or depletion, for positive or negative values, respectively. $P$-values were adjusted for multiple comparisons using Bonferroni correction. Mann-Whitney U test (two-sided) followed by Holm correction for multiple testing (when applicable) was used to assess (i) the median number of repeats difference between ataxia patients and controls; (ii) the age at onset difference between distinct patient groups; (iii) the difference in the number of triplets quantified by Nanopore/STR fragment size for samples for which STRling detected or not *FGF14* expansions; and (iv) the standard deviation of the number of triplets difference between various categories of main repeats and interruptions. The details and results of all statistical tests performed appear in the Source Data file and Supplementary Data 4.

## Circular dichroism spectroscopy

The AAG and AAGGAG repeats and the complementary counterparts were purchased as 25-mer DNA and RNA oligos from Microsynth (Switzerland) and dissolved in nuclease-free water at 100 μM. The sequences of the oligos are displayed in Fig. 7C and are identical for DNA and RNA except that thymidines were replaced by uracils in RNA. The final concentration of oligos was 50 μM in 10 mM cacodylate buffer (pH 7.2) either supplemented with 100 mM K$^+$ or Li$^+$ as indicated. Secondary structure formation was carried out by heating the oligos to 95 °C for 5 min and slowly decreasing the temperature with a ramp rate of 0.01 °C/s to 20 °C using the LightCycler® LC480II (Roche). The folded oligos were kept at 4 °C overnight and measured the next day. Circular dichroism (CD) spectra were measured over a spectral range of 200-340 nm on a Jasco J-710 CD spectropolarimeter coupled to a Jasco PFD-3505 Peltier temperature controller. All measurements were carried out at 20 °C in a quartz cuvette with a 1 mm path length using a scanning speed of 200 nm/min, a response time of 2 s, a bandwidth of 1 nm, and an accumulation of four spectra.

## Reporting summary

Further information on research design is available in the Nature Portfolio Reporting Summary linked to this article.

## Data availability

All plots displaying the structure of *FGF14* alleles generated in this study and repeat expansion catalogs (JSON files) are available in a zenodo repository (https://doi.org/10.5281/zenodo.12542853). Nanopore reads have been deposited in the European Genome-phenome Archive (EGA, http://www.ebi.ac.uk/ega), which is hosted at the EBI and the Center for Genomic Regulation (CRG), under the Study Accession Number EGAS50000000481. Individual genome data could not be made publicly available due to ethical considerations. Controlled access to human sequences is necessary to protect the privacy of participants and to ensure that the use of the data conforms to ethical and legal standards, particularly in relation to data protection laws in Germany and Europe. The sharing of individual genome data with other researchers is only possible with the approval of the ethics committee and should comply with current data protection regulations in Germany, with the consent of each participant. Requests for

access to controlled data can be sent by e-mail to the corresponding author. The timeframe for responding to access requests is usually within 4–6 weeks from the date of submission. Requests for nanopore reads will be reviewed by a data access committee and are subject to a data access policy. Requests for genome data will be reviewed by the Ethics Committee and applicants may be required to provide additional documentation to ensure compliance with the ethical framework of the study. Data use agreements will outline specific restrictions on the use of the data. Specific restrictions on the use of controlled data may include, but are not limited to, restrictions on the sharing of data with third parties, requirements that data be used only for research purposes, and an agreement that no attempts will be made to re-identify individual participants. All other data supporting the findings described in this manuscript are available in the article and its Supplementary Information files. Source data are provided with this paper.

## Code availability

Custom scripts used in this study are available on GitHub using the following links: https://github.com/kilpert/FGF14_basecalling.git; https://github.com/kilpert/FGF14_analyses.git (https://doi.org/10.5281/zenodo.12655177 and https://doi.org/10.5281/zenodo.12654084).

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

## Acknowledgements

We thank all patients, family members, and control subjects who participated in this study. We also thank Nele Bohne and Dr. Theresa Kühnel for their contributions to establishing the LR-PCR analysis and Dr. Christine Beuck for her expert technical guidance with the CD spectroscopy measurements. This work was supported by the Tom Wahlig foundation, the University Hospital Essen, and the Deutsche Forschungsgemeinschaft (DFG, German Research Foundation) Research Infrastructure West German Genome Center (project 407493903) as part of the Next Generation Sequencing Competence Network (project 423957469). Short-read genome sequencing was carried out at the production site Cologne (Cologne Center for Genomics; CCG). C.D. received the DFG 458099954 as part of the DFG Sequencing call #3. S.Kl. was supported by the German Federal Ministry of Education and Research (BMBF) through the TreatHSP consortium (01GM1905C). M.S. and -as an associated partner- D.T. and S. Kl. were also supported by the DFG (grant number 441409627), as part of the PROSPAX consortium under the frame of European Joint Program on Rare Diseases (EJP-RD), under the EJP RD COFUND-EJP N° 825575. J.P. was supported by the Clinician Scientist program PRECISE.net funded by the Else Kröner-Fresenius-Stiftung. F.J.K. received funding from the DFG Research Unit FOR2488 and LOCOTACT CRC/TR 296. Genome sequencing in Spanish patients was supported by the Undiagnosed Rare Diseases Program of Catalonia (URDCat; PERIS SLT002/16/00174) from the Autonomous Government of Catalonia; the Biomedical Research Networking Center on Rare Diseases (CIBERER, ACCI19-759); the Hesperia Foundation (Royal House of Spain), and La Marató de TV3 Foundation with project 202006–30 to C.C. and A.P. and the Instituto de Salud Carlos III co-funded by the Fondo Europeo de Desarrollo Regional (FEDER), Unión Europea, una manera de hacer Europa (FIS PI20/00758 and PI23/01090) to C.C., and IMPACT-Genomica (IMP/00009) to A.P. M.R. was funded by the Center for Biomedical Research on Rare Diseases, an initiative of the Instituto de Salud Carlos III. We thank the CERCA Program/Generalitat de Catalunya for institutional support. The license to use BioRender was obtained through University Duisburg-Essen. Open access funding enabled and organized by Projekt DEAL.

## Author contributions

L.M., F.E., E.L., G.S.H., and S.Ka. contributed to data acquisition and data analysis. L.M., E.L., and S.Ka performed the experiments. E.L., F.Ki., and C.S. performed computational analyses. C.D. conceived and supervised the study. A.T., M.St., J.P., A.S., M.R., M.M.dlP., C.C., K.B., U.R., S.P., F.J.K., M.Sy., T.W., M.A., T.B.H., P.J.L., K-H.J., A.P., S.Kl., D.T. contributed genetic data, patient data, analysis and/or critically revised the manuscript for intellectual content. C.D., F.E., and D.T. drafted the manuscript. All authors reviewed and approved the manuscript.

## Funding

## Competing interests

MS received consultancy honoraria from Solaxa. All other authors report no competing interests.

## Additional information

Lars Mohren [1,20], Friedrich Erdlenbruch [2,20], Elsa Leitão [1,20], Fabian Kilpert [1,20], G. Sebastian Hönes [3], Sabine Kaya[1], Christopher Schröder [1], Andreas Thieme [2], Marc Sturm [4], Joohyun Park[4], Agatha Schlüter[5,6], Montserrat Ruiz [5,6], Moisés Morales de la Prida [5,7], Carlos Casasnovas [5,6,7], Kerstin Becker [8], Ulla Roggenbuck[9], Sonali Pechlivanis[9,10], Frank J. Kaiser[1,11], Matthis Synofzik [12,13], Thomas Wirth [14,15,16], Mathieu Anheim[14,15,16], Tobias B. Haack [4,17], Paul J. Lockhart [18], Karl-Heinz Jöckel[9], Aurora Pujol [5,6,19], Stephan Klebe[2], Dagmar Timmann[2] & Christel Depienne [1] ✉

[1]Institute of Human Genetics, University Hospital Essen, University Duisburg-Essen, Essen, Germany. [2]Department of Neurology and Center for Translational Neuro- and Behavioral Sciences (C-TNBS), University Hospital Essen, University Duisburg-Essen, Essen, Germany. [3]Department of Endocrinology, Diabetes and Metabolism, University Hospital Essen, University of Duisburg-Essen, Essen, Germany. [4]Institute of Medical Genetics and Applied Genomics, University of Tübingen, Tübingen, Germany. [5]Neurometabolic Diseases Laboratory, Bellvitge Biomedical Research Institute (IDIBELL), Barcelona, Spain. [6]CIBERER, Centro de Investigación Biomédica en Red de Enfermedades Raras, ISCIII, Madrid, Spain. [7]Neuromuscular Unit, Neurology Department, Bellvitge University Hospital, Barcelona, Spain. [8]Cologne Center for Genomics (CCG), University of Cologne, Faculty of Medicine and University Hospital Cologne, Cologne, Germany. [9]Institute for Medical Informatics, Biometry and Epidemiology, University Hospital Essen, University of Duisburg-Essen, Essen, Germany. [10]Institute of Asthma and Allergy Prevention, Helmholtz Zentrum München, German Research Center for Environmental Health, Neuherberg, Germany. [11]Essener Zentrum für Seltene Erkrankungen (EZSE), Universitätsklinikum Essen, Essen, Germany. [12]Division Translational Genomics of Neurodegenerative Diseases, Center for Neurology & Hertie Institute for Clinical Brain Research Tübingen, Tübingen, Germany. [13]German Center for Neurodegenerative Diseases (DZNE), Tübingen, Germany. [14]Service de Neurologie, Département de Neurologie, Hôpitaux Universitaires de Strasbourg, Hôpital de Hautepierre, 1, Avenue Molière, Strasbourg, Cedex, France. [15]Institut de Génétique et de Biologie Moléculaire et Cellulaire (IGBMC), INSERM-U964/CNRS-UMR7104/Université de Strasbourg, Illkirch, France. [16]Fédération de Médecine Translationnelle de Strasbourg (FMTS), Université de Strasbourg, Strasbourg, France. [17]Centre for Rare Diseases, University of Tübingen, Tübingen, Germany. [18]Bruce Lefroy Centre, Murdoch Children's Research Institute; Department of Paediatrics, The University of Melbourne, Parkville, VIC, Australia. [19]Catalan Institution of Research and Advanced Studies (ICREA), Barcelona, Spain. [20]These authors contributed equally: Lars Mohren, Friedrich Erdlenbruch, Elsa Leitão, Fabian Kilpert. ✉e-mail: christel.depienne@uk-essen.de

