## [Peer Review File · Nature Communications]

Identification and characterisation of pathogenic and non-pathogenic FGF14 repeat expansionsREVIEWER COMMENTS

Reviewer #1 (Remarks to the Author):

The authors report on their screening for the FGF14 repeat expansion in a large cohort of ataxia patients and demonstrate a large contribution of this variant to the fraction of hitherto genetically unexplained patients. Their findings result in a better estimation of the pathogenic length for this repeat. The study is well executed and the manuscript is well written. I wish to congratulate the authors on their work.

Please find my specific comments below.

Sincerely,
Wouter De Coster

Major:

The authors perform CD spectroscopy on a 25-mer of multiple sequence compositions, sense and antisense, demonstrating differences in the secondary structure of the DNA and RNA. A 25-mer is, however much shorter than the repeats which are observed in controls and patients. Could the authors speculate to which extent their observations on structure could be generalized to longer sequences with the flanking sequence?

Minor:

The authors report an individual with two larger alleles and one shorter (lines 163-164). Was a duplication of the haplotype with the expanded allele excluded here?
The authors report on a nonsense variant identified in FGF14, with a strikingly different outcome for the father and the daughter. Is there an indication that the father has the variant somatically, passing it on as a germline variant to his daughter?

Reviewer #1 (Remarks on code availability):

The code is well written and the provided workflows would, given the input data, allow users to reproduce the results or apply this to their own data. There is however a lot of code (including scripts for generating plots) and reviewing all in detail would be too time-consuming.

Reviewer #2 (Remarks to the Author):

Mohren et al. provide an interesting study investigating a large sample of FGF14 repeat expansion carriers and controls. They address relevant questions relating to the genetic underpinnings of SCA27b, including new data on the association between repeat numbers and disease manifestation. However, most of the findings are not particularly novel, and there are relevant shortcomings that should be addressed.

Strengths of the study consist of the state-of-the-art genetic approach, including genome analyses, nanopore sequencing, and bioinformatics, which have, however, similarly performed and described previously (e.g., PMID: 36516086, 37322040, 36493768). The clinical features align with previous evidence (Downbeat nystagmus, response to 4-Aminopyridin: e.g., PMID: 38507876; Episodic ataxia in the beginning: e.g., PMID: 38150853). However, the presented study does not relevantly contribute to the phenotypic characterization of SCA27b as clinical data is rather superficial and partly not appropriately presented (see below). Relevant open questions regarding the clinical picture, such as the role of neuropathy and vestibulopathy and brain imaging abnormalities in SCA27b, are not addressed.

The analysis of the flanking regions reveals fascinating data providing new insights into molecular underpinnings of repeat stability. The authors demonstrate again that sequencing the repeat expansion is crucial, showing that non-pathogenic repeat expansions are more frequent in controls. The finding that non-GAA-expansions are not pathogenic was already suggested by previous studies (PMID: 37322040, 37565404), but the authors expand on this topic. Moreover, the data on circular dichroism spectroscopy is interesting and might represent a relevant method to distinguish between different repeat conformations.

Whether specific repeat numbers should be considered pathogenic is crucial for patient counseling and has enormous implications for SCA27b research and clinical care. However, also repeat lengths between 250 and 300, considered "borderline" and disease-causing with reduced penetrance at the moment, are with 1% surprisingly common in controls (PMID: 36516086). Thus, claiming that even shorter repeat expansions are causing the disease must be robustly reflected by the data. Here, the authors show enrichment of repeat numbers between 180 and 250 in ataxia patients vs. controls (Figure 4D), which is in line with findings from other very recently published studies (e.g., PMID: 38507876) mentioned by the authors in the discussion. The authors should emphasize this enrichment also in the results part on "Intermediate FGF14 alleles".

However, further evidence that ataxia is caused by alleles shorter than 250 repeats from family studies is relatively weak, as segregation is only demonstrated in one (!) single family. Looking at the supplement, Supplementary Figure 3A provides that not a single family has genetic data for two generations, hindering conclusions about segregation. However, the one and only family E19-0805 with convincing data on segregation in the context of an autosomal dominant inheritance is presented in Figure 3. The other family mentioned by the authors in lines 209-211 (SPG79) has two affected siblings- however, the parents, who are indicated to be still alive, are not affected, so no autosomal-dominant inheritance is demonstrated. Thus, this family does not directly support a pathogenic role of FGF14 repeat expansions in this family (and no "evidence(d) by segregation in a Spanish family" as stated by the authors) as a to date unknown recessive inheritance might also be possible in the context of the pedigree. The authors discuss these and further shortcomings but nevertheless conclude that the data presented in this study is robust enough to revise the threshold for pathogenicity to a lower allele number, which the data might not reflect. I would suggest at least to tone down this suggestion in the last paragraph of the manuscript.

The presentation of clinical data could be more convincing. Figure 6 is difficult to understand, and the relevance of the data in its presented form is questionable. First, the figure legend should mention the statistical tests used and the total number of individuals presented. There is no comment on missing data, which is obviously an issue for several categories. How was impaired sense of vibration defined (threshold?), and, more importantly, how was cognitive decline assessed? How was age at onset assessed? By patient history only? And if so, which symptoms were used to define the onset?

In general, presenting clinical data distinguishing groups according to the number of repeats, as shown in Figure 6 and the results part, does not seem to provide relevant deeper insights while reducing the sample size in each group. Moreover, investigating differences between these groups without considering disease duration might lead to misleading comparisons. In this meaning, the time between the first and last examination is not reported (similar for all participants or not?). This is particularly relevant, as due to splitting the groups, the single groups are partly very small with $n < 10$. Furthermore, showing only bars with percentages if one group is $n=1$ is inappropriate. Thus, I suggest revising Figure 6 and the presentation of clinical data (also in the results part, where age at examination and disease duration for the different groups should be provided).

Minor comments:

What does "median repeat numbers" mean if single individuals are reported (line 155)? I guess this

refers to the methodology, but this should be explained.

Were there specific clinical features of individuals with biallelic repeat expansions (lines 159-167)? These individuals might be more interesting to present as case reports than the patients reported in the supplement.

Clinical data on controls was not reported, partly because it is not available, as blood from not-at-all-phenotyped blood donors was used. However, it seems like there is also no clinical data from the Heinz Nixdorf RECALL Study participant?! If so, the fact that the control participants did not undergo at least a basic neurological examination should be mentioned at least in the limitations section, which is completely missing at the moment.

Reviewer #2 (Remarks on code availability):

NA

Reviewer #3 (Remarks to the Author):

In Manuscript#: NCOMMS-24-08210, the authors tackle a crucial question concerning the intriguing aspects of FGF14 repeat expansions, such as pathogenic thresholds and the complete sequences of pathogenic and non-pathogenic alleles. While the study possesses the necessary data to address these questions, the current structure of the manuscript significantly impedes the comprehension and appreciation of these results.

What are the noteworthy results? Targeted nanopore sequencing is presented as a cost-effective method that can be easily implemented to sequence FGF14 expansions for undiagnosed cerebellar ataxias. Utilizing the median number of repeats, instead of the mean, shows more accurate results (closer to the size observed on gel).

Will the work be of significance to the field and related fields? How does it compare to the established literature? The work is original and presents important results. However, it lacks a proper analysis of the obtained data in a deeper and more comprehensive way to show new conclusions. It would also be beneficial to remove sections that discuss topics outside the scope of the presented data, such as physiopathology, somatic variability, or new phenotypes based on comparisons with rats.

Does the work support the conclusions and claims, or is additional evidence needed? While the authors conclude, "this study reveals the complete sequence of pathogenic and non-pathogenic FGF14 expansions. We suggest that pure AAG expansions are pathogenic from a lower threshold (comprised between 180 and 220 repeats)... other hexanucleotide repeats motifs are not pathogenic," a clearer explanation of how this is demonstrated and analyzed in the discussion is missing, as it is mixed with many other concepts. This part lacks sufficient evidence, sample size, and methodology to conclude, "and accounts for at least 23% of patients with adult-onset cerebellar ataxia in European populations."

Are there any flaws in the data analysis, interpretation, and conclusions? Do these prohibit publication or require revision? As stated above, a major revision is required, especially for:

Age of onset analysis: underpowered, standard deviation not reported, and cohorts included in meta-analysis not done in a systematic and comprehensive way.

Predictive value and sensitivity: there is no explanation on ROC or sensitivity and specificity analysis being conducted.

Comparison with rat phenotypes.

Inclusion of biallelic variants.

Assessing frequency mixing with other retrospective data of cohorts that are not explained how have

been analyzed.

Addressing RFC1 Repeat Expansions: Given the manuscript's focus on undiagnosed ataxias, a discussion on RFC1 repeat expansions could provide a more comprehensive understanding of cerebellar ataxias.

In summary: While findings are original and potentially impactful, the manuscript could be improved by:

Deeper and Consistent Analysis: A more comprehensive analysis of the data could reveal nuanced insights and new conclusions. This might involve a closer examination of the pathogenicity thresholds for FGF14 expansions and their implications.

Clear Connection to Objectives: Ensure each manuscript section, from the introduction to the conclusions, clearly supports the study's main objectives.

Methodological Clarity and Statistical Analysis: Enhance explanations of methodological choices and statistical analyses to support your conclusions more robustly.

Focused Discussion: Narrow the discussion to directly relevant topics. Extraneous discussions on physiopathology, variability, or comparisons with rat phenotypes.

Reviewer #3 (Remarks on code availability):

Yes, the code provides README file. It is well explained. It even has a demo dataset

Reviewer #4 (Remarks to the Author):

This important paper provides a comprehensive characterization of FGF14 STR expansions and SCA27B. The study investigates FGF14 expansions in several patient & control cohorts and employs multiple technologies to shed light on key questions, including:

- effective approaches to diagnosing pathogenic expansions at this locus
- the accuracy of widely-used short-read genome STR callers in detecting FGF14 expansions
- penetrance, pathogenic thresholds, age of onset, phenotypic variability, disease mechanism and the roles of secondary structures in pure and interrupted repeats.

Overall, this in-depth study makes multiple highly valuable contributions to our understanding of FGF14 / SCA27B.

I have a relatively long list of questions and requests for clarification, but consider them to be secondary, and up to the authors to decide which ones they think are easy and/or important to address. Overall, I thought the study was well written, with the primary points being well supported and easy to understand.

Introduction:

- "with an incidence up to 60% of families in this population" - may be worth stating the obvious - that the 60% incidence is within a cohort of late-onset cerebellar ataxia cases rather than the Canadian population.

Results:

- "Significant values were detected for both individuals of four families for which genome data were available." - unclear what "both individuals" is referring to.
- "In addition, STRling detected an expansion of another motif (AAGGAG) at the same locus in one family from Spain." Not clear what the reader should take away from this sentence - is it that STRling can accurately distinguish pure vs. interrupted repeats.
- "was confirmed for 18 of the 26 individuals, all with cerebellar ataxia" - the previous paragraph mentioned that STRling detected expansions in 22 out of 48 ataxia cases. What does the 26 number refer to here?

- "The eight remaining individuals, including the three with another neurological disorder.." - three individuals are mentioned in the preceding paragraph. What are the other 5 non-ataxia cases? I initially thought these were the remaining 8 out of 26 (ie. 26 minus 18 from the preceding sentence), but it says those individuals all have cerebellar ataxia, while these 8 include 3 with another neurological disorder.

- "(PCR product ≥ 700 bp; triplet repeat number ≥ 180)" When I initially read the paper, I automatically divided 700bp by 3 and thought it was equivalent to 233 repeats. After reading the paper closely, I realized these were PCR fragment sizes and should not simply be divided by 3. Would it make sense to replace "700bp" and "650bp" with the equivalent triplet repeat number wherever only 1 number is mentioned (eg. Figure 1 boxes) or, alternatively, add some additional label there like "700bp PCR frag." ?

- "Of note, we also confirmed that all patients that had no outlier values detected by STRling only had small FGF14 alleles (Supplementary Fig. 1; Supplementary Data 1)." Would be helpful to reference Supplementary Fig. 1B specifically.

- "... patients negative for SCA27B (n=90; Supplementary Data 3 and 4)" - just to double-check my understanding, does this "negative" category include individuals with truncating variants? - also, "negative for SCA27B" seems like a phenotypic category, while the other groups are defined based on genotype (ie. number of repeats at the FGF14 locus). Would it make sense to instead call the "negative" category just "repeats < 180?" or where appropriate, "repeats < 180 and not p.Leu80"?

- "Frequency of SCA27B in the German cohort" .. out of curiosity, in case it's not difficult to add statistics for recessive disorders, were there also diagnoses for RFC1 and/or Friedreich's Ataxia ?

- "Taken together with the expansions detected from genome data, 49 individuals from 40 families out of 134 independent families analyzed .. had an expanded AAG allele"

I interpreted Figure 1 as saying that 59 individuals had expansions (ie. "Outliers (> 700bp) Ataxia (n=59)") - how does that reconcile with the 49 individuals mentioned here?

- "Four patients from three families had an expansion of the AAGGAG motif (Supplementary Fig. 2B). The AAGGAG expansion segregated in two affected individuals from a Spanish family but was present in only the index case but not the affected mother of a German family (Supplementary Fig. 3B) and was further considered as non-pathogenic, as already reported." I'm understanding this as 3 affected patients having AAGGAG expansions and 1 affected patient having a non-pathogenic AAGGAG expansion. If that's right, does it mean AAGGAG expansions should be considered reduced-penetrance rather than non-pathogenic?

Discussion

- "... ExpansionHunter using the STRipy interface. When used with off-target/extended mode, this tool provides a closer estimation of repeat numbers but the values cannot be fully trusted and need validation by a more reliable method." If possible, it would be nice to explicitly state the PPV and sensitivity of EH+STRipy given your recommended pathogenic thresholds (and any genotype filters).

- "... errors can usually be easily distinguished from true sequence variations if the sequencing depth is sufficient" Would it be possible to recommend a minimum ONT sequencing depth?

- "Overall, the frequency of pathogenic FGF14 expansions in the control population we studied, which is representative of the population from North Rhine-Westphalia, using 250 repeats as a threshold, is 7 out of 802 (0.87%). In particular, we observed two control individuals (46 and 70 years old) with a pure AAG repeat expansion ≥ 300 (313 and 319 repeats, respectively)." Would it be possible to provide separate penetrance estimates for pure vs. interrupted expansions? I think this could help analysts with interpretation of cases in our rare disease cohorts.

- "Additional causes include a pathogenic SCA6 expansion." Not immediately clear if this should be read as "SCA6 expansions were found in some of our cases" or "SCA6 is another possible cause of ataxia"

- "... size of the smaller allele .. play an additive effect and partially determine the penetrance of alleles <320 repeats." Out of curiosity, was there any indication that individuals with biallelic expansions (regardless of their AAO) had faster disease progression once symptoms began (which would be consistent with the model from [Kaplan,Shapiro 2007])?

- Do you have any recommendations for whether the pre-repeat sequence (as determined by short

read sequencing) should be looked at during analysis/diagnosis? If yes, how should this be included in decision making?

Figure 2:

What does "VAR/+" mean?

What do the "2", "3", "4", "6" labels on some individuals mean?

Caption: Should "Black symbols indicates subjects examined and sampled in the study" be "Black symbols *indicate affected* subjects examined and sampled in the study"?

Figure 3:

3B: it's hard to distinguish the red color for AGG from the maroon color for ACG. AGG is the same motif as GAG (yellow) just shifted by 1bp, while ACG is a very different motif. It's currently unclear whether the colors on the red/orange colors on the M95411 and M81457 plots should be interpreted as a mixture of GAA/GAG or GAA/GAC motifs. A farther-from-red color (green?) should clarify this. Also it would be nice if the pathogenic threshold could be placed on these graphs as a vertical line and/or the x-axis labeled as "(bp)" (so it's clear that it's not the # of repeats).

3B: would it make sense to explicitly state in the caption that the pink AAC streaks are likely sequencing errors (as suggested in the Discussion)?

3C: E20-0199 shows an affected individual with a pathogenic-range expansion that's full of interruptions. Is this not meant to be taken as evidence that AAGGAG interruptions are still pathogenic (but have reduced penetrance relative to pure repeats)?

3D: wow, impressive accuracy

3E: It would be helpful to add a vertical line for the pathogenic threshold. So many interesting things in this plot. Can more be said about 1) the single outlier orange dot way above the diagonal? could its read visualizations be shown? Also, it's initially surprising that the orange dots all appear to be clustered together around 1000 bp - is this just that elsewhere they're hidden below the blue dots? The main question this plot raises for me is - why do interrupted repeats appear to show the same (or more) somatic instability compared to pure repeats? I thought a core property of STRs is that interruptions significantly reduce the mutation rate. Is there some way to show that the standard deviation metric is primarily measuring underlying somatic variation and is not a technical artifact of ONT sequencing in long repeats / does not have a high false-positive rate?

Figure 4:

4A: can more be said about the interruption patterns referred to as "AAG interruption" and "AAG interruption?" - ideally by providing example read visualizations (or just more detailed descriptions). Currently, it's not clear how to recognize these in our own data, and what the implications are if we do see them. Figure 5 lists single interruptions in detail, but that's further down in the paper, and it's unclear how the "interruption", "interruption?" labels in this figure map to the list in 5A.

4B: the caption uses "n" for the # of individuals, but the plot labels use "n" as allele counts. Might be clearer to put "n = 269 alleles" ..

4D: what is the gray "ns" category in the legend? I don't see it in the plot. Also, would it be possible to have 2 plots like this - one for pure AAG (or all) alleles and one for AAGGAG alleles?

Figure 5:

5A: what is the "Additional repeat" list? My guess is that these are interspersed within the repeat sequences of some samples, but are rarer than AAGGAG main repeats (and possibly cover less of the repeat sequence?). Would it make sense to combine these with the "single interruption" list into one "scattered interruptions" list?

Figure 6:

If possible, I think it would be helpful to show stats for categories where 0 individuals are affected with the given phenotype (for example, the expansion-negative category for Migraine, or the $180 \leq \text{exp} < 250$ for Seizures/epilepsy). Also, out of curiosity, what does it mean that impaired gait, impaired stand, and dysarthria appear to be non-correlated or anti-correlated with expansion size?

Finally, in case there are sufficient numbers of individuals for some categories, it would be interesting to see how these phenotypes are associated with interrupted vs. pure expansions.

Figure 7:

7A: The caption "Comparison of the AAO in patients negative for SCA27A/SCA27B.." seems contradictory with the figure title "progression in patients with SCA27B/SCA27A." I had assumed the "negative" category on the x-axis meant patients that don't have FGF14 expansions or known SCA27A variants.

7F, 7G: a brief summary of what SARA and ICARS scores measure would be appreciated - along with any discussion of why scores appear to be more stable for individuals with > 300 repeats than those with $250 \leq x < 300$

Sup. Figure 1:

if not difficult to add, it would be interesting if colors could be used also in this figure to show which dots represent interrupted vs. pure repeats.

Sup. Figure 5:

5A: in these box plots, do the dots represent single nanopore reads?

5D: same comment as for figure 4A: I find the "AAG interruption" vs. "AAG interruption?" labels to be confusing. For example, which labels would apply to the interruption patterns shown in patients 998894 and 99855 in Sup. Figure 4?

Sup. Figure 7:

7C: Caption doesn't say what "Fam (IP)" and "Spo" stand for.

Case reports:

M83382;#1: sentence with missing phrase: "Genome sequencing including identified a heterozygous splice ..."

Methods - Targeted sequencing with Oxford Nanopore Technologies (ONT):

"We considered only reads containing 2x AAG (i.e. AAGAAG)." not sure what this means.

Reviewer #5 (Remarks to the Author):

Mohren et al. discuss SCA27B associated AAG/GAA repeat expansion in the intronic region of FGF14 in terms of the repeat number, average onset age, influence of pre-repeat regions and interruptions.

Although the authors have put lots of effort in terms of the sample size (both the control and patients), I am not quite convinced about the novelty of the outcome discussed here. As mentioned by the authors themselves in the abstract, the association between the GAA/AAG repeat and SCA27B is already known. This study investigates in detail about the average onset age, influence of pre-repeat regions and interruptions, which is a minimal advancement in understanding the mechanism of SCA27B. Although I don't deny the importance of the results presented here, I don't think this work is suitable for Nature communications.

Minor:

- 1) the authors should consider mentioning AAG/GAA expansion in Friedreich's ataxia in the manuscript
- 2) line #499: "bioinformatic tool"  how does it differ from SATRling?
- 3) How have the authors verified the quality of the reads?
- 4) Table 1 and Figure 6  why should there be an AAO for negative?

RESPONSES TO REVIEWERS

We thank the reviewers for their valuable comments and overall positive assessment of our manuscript. Based on the feedback received, we understood that two main issues needed to be addressed.

First, the novelty of the findings presented was not immediately clear.

We have now rephrased the abstract and parts of the main text to distinguish our work from previous studies mentioned by the reviewers.

Second, our manuscript was noted to lack clarity in its presentation, as evidenced by certain points raised by reviewers for which the information was already available but may not have been sufficiently visible or clear. We have worked on improving this aspect by modifying the text and figures to enhance clarity.

Moreover, we would like to highlight that we included additional patient data (14 new patients with ataxia, one affected relative of a previous family and four unaffected family relatives of a family with intermediate alleles). The changes appear in red in our manuscript. In addition, the responses to the reviewers' comments are listed, point by point, below.

Reviewer #1:

The authors report on their screening for the FGF14 repeat expansion in a large cohort of ataxia patients and demonstrate a large contribution of this variant to the fraction of hitherto genetically unexplained patients. Their findings result in a better estimation of the pathogenic length for this repeat. The study is well executed and the manuscript is well written. I wish to congratulate the authors on their work.

Please find my specific comments below.

Sincerely,
Wouter De Coster

We warmly thank the reviewer for his positive and thoughtful assessment of our manuscript.

1-Major: The authors perform CD spectroscopy on a 25-mer of multiple sequence compositions, sense and antisense, demonstrating differences in the secondary structure of the DNA and RNA. A 25-mer is, however much shorter than the repeats which are observed in controls and patients. Could the authors speculate to which extent their observations on structure could be generalized to longer sequences with the flanking sequence?

The 25-mer oligonucleotides used for CD spectroscopy are indeed much shorter than the observed repeats. However, employing short sequences for basic structural analysis through CD spectroscopy remains the current standard practice (doi:10.1093/nar/gkp026). For instance, the quadruplex structure of the telomeric repeat TTAGGG is detectable in a 25-mer. Longer repeat lengths up to 96 bp confirmed the quadruplex formation and revealed a "bead-on-a-string" conformation with several consecutive quadruplex structures (doi: 10.1093/nar/gkaa1285). One major advantage of shorter sequences is the reduction in complexity as longer sequences can form complex higher-order structures with, e.g., hairpins and quadruplexes, that can result in an overlay of different spectra and thus interfere with one another (doi.org/10.3390/biom11111617).

With regard to the flanking region, we cannot exclude that this might have an effect on the structure of the adjacent repeat sequence. However, this should theoretically affect the extremity of the repeat. To avoid any spectral interference caused by flanking sequences, we only tested the isolated repeat sequence to get an insight into the structure of the repeat sequence.

Whether our observation can be extended to longer sequences and whether predicted structures are able to form *in vivo* remains speculative and would require deeper analysis, as the secondary structure formation of such nucleic acid conformations depends on several factors such as pH, presence or absence of cofactors, ionic strength, interaction with DNA/RNA binding proteins, intracellular concentrations (i.e., molecular crowding), to name just a few (doi.org/10.1002/0471142700.nc0711s11, doi.org/10.1016/j.ijbiomac.2023.124442).

In this context, our main aim was to use the structural data to generate a hypothesis to be tested in a more comprehensive investigation (see discussion on page 17).

2-Minor: The authors report an individual with two larger alleles and one shorter (lines 163-164). Was a duplication of the haplotype with the expanded allele excluded here?

We agree that the observation of three alleles (one unexpanded and two expanded) could also be the result of a duplication of the region. We now provide evidence that this patient only has two copies of the *FGF14* gene, as demonstrated by two different qPCR assays investigating the first two alternative exons of the gene, on either side of the expansion (page 7, Supplementary Data 3). This provides further evidence that the two expanded alleles are indeed the result of somatic mosaicism.

3-The authors report on a nonsense variant identified in *FGF14*, with a strikingly different outcome for the father and the daughter. Is there an indication that the father has the variant somatically, passing it on as a germline variant to his daughter?

We appreciate the reviewer's question. Unfortunately, the paternal grandparents (parents of the affected father) are deceased and therefore not available for genetic analysis. In the absence of a family history in previous generations, it is possible that this variant may have arisen *de novo* during the father's development, being present in only a fraction of his cells (i.e. mosaic). Nevertheless, in reviewing the clinical characteristics of patients with *FGF14* truncating variants (Supplementary Data 5), we observed a clear variability in age at onset (ranging from 2 to 47 years) and clinical presentation among affected individuals. Therefore, it remains plausible that this variant is associated with variable expressivity, which is common in many other dominant neurological disorders (e.g. *SPG4*, *ITPR1*, etc.).

4-Remarks on code availability:

The code is well written and the provided workflows would, given the input data, allow users to reproduce the results or apply this to their own data. There is however a lot of code (including scripts for generating plots) and reviewing all in detail would be too time-consuming.

We thank the reviewer for his assessment of the availability of the code. We make it a priority that the code developed for this work can be used by other groups interested in sequencing *FGF14* expansions.

Reviewer #2:

5-Mohren et al. provide an interesting study investigating a large sample of *FGF14* repeat expansion carriers and controls. They address relevant questions relating to the genetic underpinnings of

SCA27b, including new data on the association between repeat numbers and disease manifestation. However, most of the findings are not particularly novel, and there are relevant shortcomings that should be addressed.

Strengths of the study consist of the state-of-the-art genetic approach, including genome analyses, nanopore sequencing, and bioinformatics, which have, however, similarly performed and described previously (e.g., PMID: 36516086, 37322040, 36493768).

To the best of our knowledge, there has been no systematic evaluation of the performance of bioinformatics tools specifically designed to detect repeat expansions from short-read genomic data in these studies. It is worth noting that the studies cited above used ExpansionHunter DeNovo, whereas we used STRling, a newer tool that provided better performance in our hands (as detailed in the Methods section). Furthermore, the performance (predictive value and sensitivity) of any tool for diagnosing pathogenic *FGF14* expansions at the individual level have not been evaluated. Given that short-read data is currently the predominant method for diagnosing patients with rare diseases, this analysis, together with the development of an improved, cost-effective method for amplifying and sequencing *FGF14* expansions, is of significant importance to the scientific community.

Furthermore, none of the previous studies compared the sequence of *FGF14* repeat expansions between patients and controls. Our study provides an integrative and complete analysis of *FGF14* alleles in large, unpublished cohorts of patients and controls, and demonstrates for the first time the relevance of integrating flanking regions and interruptions as possible modifiers of SCA27B disease expression and penetrance. It should be noted that most of the previously known findings (molecular, clinical, nature of pathogenic versus non-pathogenic expansions) were scattered in different publications and some of the findings (non-pathogenicity of AAGGAG expansions, lower thresholds of pure expansions) were suggested by single observations but not statistically proven. For these reasons, we believe that our work adds substantial knowledge to previous efforts.

6-The clinical features align with previous evidence (Downbeat nystagmus, response to 4-Aminopyridin: e.g., PMID: 38507876; Episodic ataxia in the beginning: e.g., PMID: 38150853). However, the presented study does not relevantly contribute to the phenotypic characterization of SCA27b as clinical data is rather superficial and partly not appropriately presented (see below).

We agree with the reviewer's observation that the clinical manifestations described in our study are consistent with previous observations. However, the primary objective in the clinical section of our manuscript is to actually demonstrate that SCA27B is a clinically recognizable and distinct entity regardless of the number of *FGF14* repeats - a finding that represents a novel contribution to the field.

Previous evidence has suggested incomplete penetrance of repeat expansions in the range of 250-300 repeats, and a recent study has even questioned the pathogenicity of repeat expansions below 300 repeats based on a comparison of repeat expansion size without including information on their sequence (Méreux et al, 2024, PMID: 38150853, now added in the discussion). Our study provides clear evidence that repeat expansions in the range of 250-300 (and even below 250) are pathogenic, with a clinical picture indistinguishable from expansions > 300 repeats but distinct from *FGF14*-negative ataxias. For the first time, we provide statistical evidence for this finding based on the analysis of a large control population that was assessed not only for size but also for sequence content.

7-Relevant open questions regarding the clinical picture, such as the role of neuropathy and vestibulopathy and brain imaging abnormalities in SCA27b, are not addressed.

These points represent topics that potentially require their own dedicated study and are thus outside the scope of our manuscript. For example, the study on brain imaging has been performed by another group and is available as a preprint (<https://www.medrxiv.org/content/10.1101/2024.02.16.24302945v1>).

8-The analysis of the flanking regions reveals fascinating data providing new insights into molecular underpinnings of repeat stability. The authors demonstrate again that sequencing the repeat expansion is crucial, showing that non-pathogenic repeat expansions are more frequent in controls. The finding that non-GAA-expansions are not pathogenic was already suggested by previous studies (PMID: 37322040, 37565404), but the authors expand on this topic. Moreover, the data on circular dichroism spectroscopy is interesting and might represent a relevant method to distinguish between different repeat conformations.

We agree, and thank the reviewer for this comment.

9-Whether specific repeat numbers should be considered pathogenic is crucial for patient counseling and has enormous implications for SCA27b research and clinical care. However, also repeat lengths between 250 and 300, considered “borderline” and disease-causing with reduced penetrance at the moment, are with 1% surprisingly common in controls (PMID: 36516086). Thus, claiming that even shorter repeat expansions are causing the disease must be robustly reflected by the data. Here, the authors show enrichment of repeat numbers between 180 and 250 in ataxia patients vs. controls (Figure 4D), which is in line with findings from other very recently published studies (e.g., PMID: 38507876) mentioned by the authors in the discussion. The authors should emphasize this enrichment also in the results part on “Intermediate FGF14 alleles”.

We rephrased the part on intermediate alleles to clarify the results and added a panel of figure 4D to better display the difference observed between patients and controls for values above 180. Additionally, the Pellerin et al. reference (PMID: 38507876), published in Ebiomedicine during the review of our manuscript, has now been updated.

10-However, further evidence that ataxia is caused by alleles shorter than 250 repeats from family studies is relatively weak, as segregation is only demonstrated in one (!) single family. Looking at the supplement, Supplementary Figure 3A provides that not a single family has genetic data for two generations, hindering conclusions about segregation. However, the one and only family E19-0805 with convincing data on segregation in the context of an autosomal dominant inheritance is presented in Figure 3. The other family mentioned by the authors in lines 209-211 (SPG79) has two affected siblings— however, the parents, who are indicated to be still alive, are not affected, so no autosomal-dominant inheritance is demonstrated. Thus, this family does not directly support a pathogenic role of FGF14 repeat expansions in this family (and no “evidence(d) by segregation in a Spanish family” as stated by the authors) as a to date unknown recessive inheritance might also be possible in the context of the pedigree. The authors discuss these and further shortcomings but nevertheless conclude that the data presented in this study is robust enough to revise the threshold for pathogenicity to a lower allele number, which the data might not reflect. I would suggest at least to tone down this suggestion in the last paragraph of the manuscript.

Our study differs from family-driven research discoveries such as PMID: 36516086 and is designed as a prospective research study of repeat expansions in exome-negative individuals. Because of this setting, our study reflects the situation faced in clinics where segregation of the expansion is not always available for family members due to geographical distance of living relatives, small family size or simply the unavailability of unaffected family members for research studies. To overcome this

difficulty, we analyzed a very large geographically matched control population and showed statistical significance for enrichment of *FGF14* alleles below the current threshold of 250, demonstrating for the first time that alleles below the current threshold are more frequent in cases than controls and are therefore associated with an increased risk of ataxia. As intermediate alleles are more likely to act as susceptibility factors than penetrant pathogenic alleles, segregation of expansion within families, used to assess the pathogenicity of alleles in rare monogenic disorders, is no longer a criterion to exclude the involvement of risk factors, especially since, as the reviewer point out, the frequency of *FGF14* expansions in control populations (0.8% in the population we analyzed) is actually higher than the incidence commonly used to define rare disorders (1 in 2,000 individuals).

Nevertheless, we recently had the opportunity to extend segregation analysis for family SPG-79 and these results now appear in Supplementary Figure 3B. The results show that the expanded allele present both affected male siblings are retracted upon transmission to their unaffected offspring, a result compatible with the expression of the disease only in these two members of this family.

11-The presentation of clinical data could be more convincing. Figure 6 is difficult to understand, and the relevance of the data in its presented form is questionable. First, the figure legend should mention the statistical tests used and the total number of individuals presented. There is no comment on missing data, which is obviously an issue for several categories.

Based on this comment and comment 13, we have removed the previous Figure 6 and now present the clinical data as Table 1.

The statistical tests used are indicated in the Table legend and Supplementary Data 6.

We have added a sentence in the Methods (in the new paragraph “clinical evaluation”, page 18) on how we dealt with missing data.

12-How was impaired sense of vibration defined (threshold?) and, more importantly, how was cognitive decline assessed? How was age at onset assessed? By patient history only? And if so, which symptoms were used to define the onset?

Impaired sense of vibration (reduced pallesthesia) was defined as $\leq 5/8$ on Rydel–Seiffer tuning fork at the medial malleolus (same threshold as used in Wilke et al., 2023).

Cognitive decline was assessed based on medical history and clinical judgement. It was documented as part of the Inventory of Non-Ataxia Signs (INAS).

The age at onset of the disease was assessed based on the medical history. The first symptom reported by the patient was used as age at onset, particularly, first episodic occurrence of gait ataxia, oscillopsia (as indication of downbeat nystagmus) or ataxia of gait.

This information is now included in a new paragraph “clinical evaluation” in the methods together with corresponding references.

13-In general, presenting clinical data distinguishing groups according to the number of repeats, as shown in Figure 6 and the results part, does not seem to provide relevant deeper insights while reducing the sample size in each group. Moreover, investigating differences between these groups without considering disease duration might lead to misleading comparisons. In this meaning, the time between the first and last examination is not reported (similar for all participants or not?). This is particularly relevant, as due to splitting the groups, the single groups are partly very small with $n < 10$. Furthermore, showing only bars with percentages if one group is $n=1$ is inappropriate. Thus, I suggest revising Figure 6 and the presentation of clinical data (also in the results part, where age at examination and disease duration for the different groups should be provided).

As already mentioned in response to point 6 (reviewer 2), the main objective of the clinical analysis included in our manuscript was to prove that SCA27B is a clinically recognizable and distinct entity, independent of the number of *FGF14* repeats, and investigate the possibility that repeat numbers lower than the current pathogenic threshold can also be pathogenic. We also confirm by comparing phenotypes of patients with different expansion ranges that pure expansions in the range of 250-300 repeats, currently considered to be associated with incomplete penetrance, are pathogenic with the same clinical picture and that incomplete penetrance also exists above 300 repeats. We think that together the clinical part included in this manuscript nicely complement and support the molecular results.

However, to address the concern raised by two of the reviewers, we have revised the presentation of the clinical findings. These results are now summarized in Table 1, which also includes the age at examination and disease duration. The statistical comparisons made in this table are based on the SCA27B-positive (i.e., $FGF14 \geq 250$ repeats) and SCA27B-negative groups. The breakdown of the clinical features into subgroups (numbers in all subgroups are > 10) is used to show the consistency of the phenotype across different repeat size.

Regarding the family with the nonsense variant, we have detailed clinical details for the index case only but we include the age at onset (AAO) for the two affected family members (Table 1). The only plot that remains in the revised manuscript including this family is the AAO (Fig. 6C).

Furthermore, as part of the revision, we have incorporated data from 15 additional patients, systematically analyzed during the manuscript's review, which allows to increase the numbers of patients considered in each subgroup.

Please also refer to our response to point 11.

Minor comments:

14-What does “median repeat numbers” mean if single individuals are reported (line 155)? I guess this refers to the methodology, but this should be explained.

The median repeat number refers to the way in which the repeat number is calculated from nanopore reads of the same individual containing different repeat numbers. We decided to use the median which is a more robust statistic against possible sequencing outliers.

Details about the calculation of median repeat number is available in the methods at the end of the paragraph “Targeted sequencing with Oxford Nanopore Technologies (ONT)” page 21. We have now added the reference to the methods section the first time we use the term “median repeat number”.

15-Were there specific clinical features of individuals with biallelic repeat expansions (lines 159-167)? These individuals might be more interesting to present as case reports than the patients reported in the supplement.

We now provide case reports for the six individuals with biallelic expansions included in the revised manuscript. Three out of 6 (M97638:222/278, M83382:196/284, M80332:204/311) were previously in the group considered as ‘clinical outliers’ with a more rapid progression. The three remaining patients (M96652: 165/272, M83825: 196/319 and M100781:684/498) show a progression comparable to that of patients with monoallelic expansion. Interestingly, patient M100781, who has

been newly added to the study is the only one from his family to be affected and he has two biallelic pathogenic expansions with the same putative 5' interruption (Supplementary Fig. 4A), suggesting a lower penetrance of this expansion configuration and possible recessive inheritance in this family from Iraq. This information was added in the results, discussion and supplementary material (case reports).

16-Clinical data on controls was not reported, partly because it is not available, as blood from not-at-all-phenotyped blood donors was used. However, it seems like there is also no clinical data from the Heinz Nixdorf RECALL Study participant?! If so, the fact that the control participants did not undergo at least a basic neurological examination should be mentioned at least in the limitations section, which is completely missing at the moment.

We have included details regarding the unavailability of neurological status for individuals from the HNR study in the methods (page 18) and also discuss this (page 15).

17-Remarks on code availability: NA

Reviewer #3:

18-In Manuscript#: NCOMMS-24-08210, the authors tackle a crucial question concerning the intriguing aspects of FGF14 repeat expansions, such as pathogenic thresholds and the complete sequences of pathogenic and non-pathogenic alleles. While the study possesses the necessary data to address these questions, the current structure of the manuscript significantly impedes the comprehension and appreciation of these results.

We have revised sections of the manuscript to address this concern, aiming to enhance readability and comprehension. We hope that the current version of the text is more accessible and clear.

19-What are the noteworthy results? Targeted nanopore sequencing is presented as a cost-effective method that can be easily implemented to sequence FGF14 expansions for undiagnosed cerebellar ataxias. Utilizing the median number of repeats, instead of the mean, shows more accurate results (closer to the size observed on gel).

A sentence already highlights these points in the discussion (page 14).

20-Will the work be of significance to the field and related fields? How does it compare to the established literature? The work is original and presents important results. However, it lacks a proper analysis of the obtained data in a deeper and more comprehensive way to show new conclusions. It would also be beneficial to remove sections that discuss topics outside the scope of the presented data, such as physiopathology, somatic variability, or new phenotypes based on comparisons with rats.

We have revised the text and the presentation of data in the results section to enhance comprehensiveness.

We have now removed the sentence "We found that episodic symptoms are not restricted to ataxia, but can also include episodic choreiform movements, as described in Fgf14 knock-out mice³⁹" from the discussion as suggested by the reviewer.

However, we believe that topics such as somatic variability and possible pathogenic mechanisms, which are only addressed in the Discussion section to clarify or extend our own observations, should

remain. Somatic variability of *FGF14* expansions is one of the novel findings of our study that needs to be correlated with the sequence of expansions (i.e., pure AAG versus interrupted expansions) and is therefore, in our opinion, a key topic to discuss. We also clarify some points about the pathophysiology that we think are important to mention, as these points have been eluded in previous studies: for example, the comparison of SCA27A and SCA27B suggests that the pathogenic mechanism is similar and both result from loss of function of isoform 2, which was not clear as point mutations theoretically affect both isoforms 1 and 2.

21-Does the work support the conclusions and claims, or is additional evidence needed? While the authors conclude, "this study reveals the complete sequence of pathogenic and non-pathogenic *FGF14* expansions. We suggest that pure AAG expansions are pathogenic from a lower threshold (comprised between 180 and 220 repeats)... other hexanucleotide repeats motifs are not pathogenic," a clearer explanation of how this is demonstrated and analyzed in the discussion is missing, as it is mixed with many other concepts. This part lacks sufficient evidence, sample size, and methodology to conclude, "and accounts for at least 23% of patients with adult-onset cerebellar ataxia in European populations."

In contrast to prior studies that overlooked aspects such as expansion sequence, interruptions, pre-repeat, and flanking regions, our conclusions stem from a systematic analysis and comparison of large patient and control cohorts, using an improved and uniform methodology. Our conclusions are exclusively drawn from statistically significant findings (the excess of expansions in the range of 180-249 is statistically significant). The addition of 14 new families has in particular allowed to strengthen the statistical significance of observations made for intermediate alleles.

All data from our study are accessible in supplementary Data or Source Data files, a level of transparency not achieved in any previous research on SCA27B. Parts of the results and discussion have been modified to reflect more accurately the results obtained in this study.

22-Are there any flaws in the data analysis, interpretation, and conclusions? Do these prohibit publication or require revision? As stated above, a major revision is required, especially for:
-Age of onset analysis: underpowered, standard deviation not reported, and cohorts included in meta-analysis not done in a systematic and comprehensive way.

We did perform the metaanalysis in a systematic manner (see supplementary Data 5). However, the limitations came from the limited availability of required information in previous studies. We could only include studies that provide age at onset together with a reliable estimation of the *FGF14* repeat number.

In the revised version, we included 14 new patients with ataxia, which improved the statistical power of the metaanalysis and resulted in a significant difference of the AAO between patients with 250-299 repeats and patients with ≥ 300 repeats (see page 12 and revised Figure 6C).

All graphs comparing AAO values are box plots. Box plot elements are defined as follows: center line: median; box limits: upper and lower quartiles; whiskers: 1.5 \times interquartile range; points: outliers. This information now appears in all figure legends showing boxplots. We do not calculate standard deviation because it assumes that the distribution of AAO is normal, which it is not as can be seen from the boxplots.

23-Predictive value and sensitivity: there is no explanation on ROC or sensitivity and specificity analysis being conducted.

We now include a supplementary table (Supplementary Data 8) indicating predictive values and sensitivity of the tests used to detect repeat expansions. Please see also response to point 39.

24-Comparison with rat phenotypes.

We removed the sentence. Please also see response to point 20.

25-Inclusion of biallelic variants.

We now discuss patients with biallelic expansions extensively and have added the six case reports. Please see also response to point 15.

26-Assessing frequency mixing with other retrospective data of cohorts that are not explained how have been analyzed.

The reviewer's point is somewhat unclear due to the lack of context and incomplete sentence structure. If the reviewer is referring to the frequency of other ataxia subtypes, the data are actually comparable, as they are sourced from a single cohort followed in the same institute.

27-Addressing RFC1 Repeat Expansions: Given the manuscript's focus on undiagnosed ataxias, a discussion on RFC1 repeat expansions could provide a more comprehensive understanding of cerebellar ataxias.

We have now added the frequency of *RFC1* expansions to compare them with that of *FGF14* expansions (results, pages 12-13).

28-In summary: While findings are original and potentially impactful, the manuscript could be improved by:

Deeper and Consistent Analysis: A more comprehensive analysis of the data could reveal nuanced insights and new conclusions. This might involve a closer examination of the pathogenicity thresholds for *FGF14* expansions and their implications.

Clear Connection to Objectives: Ensure each manuscript section, from the introduction to the conclusions, clearly supports the study's main objectives.

Methodological Clarity and Statistical Analysis: Enhance explanations of methodological choices and statistical analyses to support your conclusions more robustly.

Focused Discussion: Narrow the discussion to directly relevant topics. Extraneous discussions on physiopathology, variability, or comparisons with rat phenotypes.

We hope that the changes made in the revised manuscript address the reviewer's concerns.

29-Remarks on code availability: Yes, the code provides README file. It is well explained. It even has a demo dataset

We thank the reviewer for the positive evaluation of the code availability.

Reviewer #4:

30-This important paper provides a comprehensive characterization of *FGF14* STR expansions and

SCA27B. The study investigates FGF14 expansions in several patient & control cohorts and employs multiple technologies to shed light on key questions, including:

- effective approaches to diagnosing pathogenic expansions at this locus
- the accuracy of widely-used short-read genome STR callers in detecting FGF14 expansions
- penetrance, pathogenic thresholds, age of onset, phenotypic variability, disease mechanism and the roles of secondary structures in pure and interrupted repeats.

Overall, this in-depth study makes multiple highly valuable contributions to our understanding of FGF14 / SCA27B.

I have a relatively long list of questions and requests for clarification, but consider them to be secondary, and up to the authors to decide which ones they think are easy and/or important to address. Overall, I thought the study was well written, with the primary points being well supported and easy to understand.

We sincerely thank the reviewer for such a thorough assessment of our manuscript. Many of the points raised below have helped to improve the quality of the results presented.

-Introduction:

31- "with an incidence up to 60% of families in this population" - may be worth stating the obvious - that the 60% incidence is within a cohort of late-onset cerebellar ataxia cases rather than the Canadian population.

We modified the corresponding sentence (page 4).

32-Results:

- "Significant values were detected for both individuals of four families for which genome data were available." - unclear what "both individuals" is referring to.

- "In addition, STRling detected an expansion of another motif (AAGGAG) at the same locus in one family from Spain." Not clear what the reader should take away from this sentence - is it that STRling can accurately distinguish pure vs. interrupted repeats.

"was confirmed for 18 of the 26 individuals, all with cerebellar ataxia" - the previous paragraph mentioned that STRling detected expansions in 22 out of 48 ataxia cases. What does the 26 number refer to here?

- "The eight remaining individuals, including the three with another neurological disorder..." - three individuals are mentioned in the preceding paragraph. What are the other 5 non-ataxia cases?

I initially thought these were the remaining 8 out of 26 (ie. 26 minus 18 from the preceding sentence), but it says those individuals all have cerebellar ataxia, while these 8 include 3 with another neurological disorder.

We have rephrased the two first paragraphs of the results (pages 5 and 6) to clarify these points. In particular, we have added a sentence at the end of the first paragraph to clarify who are the 26 individuals (22 AAG/ataxia + 3 AAG/non-ataxia + 1 AAGGAG). In the latter individual, STRling indeed detects an expansion of an hexamer motif i.e., (AAGGAG)_{exp}. This expansion is actually large and we do not know based on this single individual if smaller expansions of this motif would also be detected.

33- "(PCR product ≥ 700 bp; triplet repeat number ≥ 180)" When I initially read the paper, I automatically divided 700bp by 3 and thought it was equivalent to 233 repeats. After reading the paper closely, I realized these were PCR fragment sizes and should not simply be divided by 3. Would it make sense to replace "700bp" and "650bp" with the equivalent triplet repeat number wherever

only 1 number is mentioned (eg. Figure 1 boxes) or, alternatively, add some additional label there like "700bp PCR frag."

We would like to keep the PCR fragment size as this is the cut-off we used in our analysis (repeat number is theoretically calculated from PCR size but the content can only be revealed by sequencing). We have now modified the corresponding sentence to refer to the methods.

34 - "Of note, we also confirmed that all patients that had no outlier values detected by STRling only had small FGF14 alleles (Supplementary Fig. 1; Supplementary Data 1)." Would be helpful to reference Supplementary Fig. 1B specifically.

We now specifically refer to Supplementary Fig. 1A (former supplementary Fig. 1B).

35- "... patients negative for SCA27B (n=90; Supplementary Data 3 and 4)" - just to double-check my understanding, does this "negative" category include individuals with truncating variants? - also, "negative for SCA27B" seems like a phenotypic category, while the other groups are defined based on genotype (ie. number of repeats at the FGF14 locus). Would it make sense to instead call the "negative" category just "repeats < 180?" or where appropriate, "repeats < 180 and not p.Leu80"?

We checked that all individuals with available genome data are negative for point variants in *FGF14* but individuals without genome data screened for repeat expansions have not been systematically analyzed for point variants in *FGF14*. Most of these individuals have had an exome analysis before being included in our research study (see methods) and should therefore be negative for point variants in *SCA27A*/point variants in *FGF14* but as we could not access all data previously generated in a diagnostic context to make sure that the entire *FGF14* gene was analyzed, we think this is more accurate to indicate this category as "SCA27B-negative patients". A paragraph with this information was added in the methods (page 19).

36- "Frequency of SCA27B in the German cohort" .. out of curiosity, in case it's not difficult to add statistics for recessive disorders, were there also diagnoses for RFC1 and/or Friedreich's Ataxia ?

We have now added the frequency of expansions on *RFC1* (CANVAS) and *FXN* (Friedreich's Ataxia) and other recessive ataxia subtypes in this cohort.

37- "Taken together with the expansions detected from genome data, 49 individuals from 40 families out of 134 independent families analyzed .. had an expanded AAG allele"
I interpreted Figure 1 as saying that 59 individuals had expansions (ie. "Outliers (> 700bp) Ataxia (n=59)") - how does that reconcile with the 49 individuals mentioned here?

Thank you for pointing out this discrepancy. The selection of individuals for sequencing was based on the observed size of the LR-PCR amplicon on the agarose gel and fragment size analysis. Indeed, both analyses are necessary to identify outliers, as fragment analysis cannot detect PCR amplicons larger than 700-1000 bp. Although fragment analysis is usually accurate, it relies on comparing peaks with a standard, which sometimes results in shifted values in some runs.

For these reasons, there were discrepancies or uncertainties about the actual size of the PCR product in some patients. Consequently, we sequenced all LR-PCR amplicons potentially greater than 700 bp based on either analysis. Fragment analysis indicated values greater than 700 bp for only 49 individuals but we sequenced a total of 59 individuals. Figure 1 has been revised to reflect this difference, including the new patients added to the manuscript during revision, resulting in a total of 67 patients with ataxia now sequenced.

38- "Four patients from three families had an expansion of the AAGGAG motif (Supplementary Fig. 2B). The AAGGAG expansion segregated in two affected individuals from a Spanish family but was present in only the index case but not the affected mother of a German family (Supplementary Fig. 3B) and was further considered as non-pathogenic, as already reported." I'm understanding this as 3 affected patients having AAGGAG expansions and 1 affected patient having a non-pathogenic AAGGAG expansion. If that's right, does it mean AAGGAG expansions should be considered reduced-penetrance rather than non-pathogenic?

We have now moved this part of the sentence to the section comparing patients and controls (page 7). The new comparisons show that there is no difference in the frequency of AAGGAG expansions between patients and controls

Discussion

39- "... ExpansionHunter using the STRipy interface. When used with off-target/extended mode, this tool provides a closer estimation of repeat numbers but the values cannot be fully trusted and need validation by a more reliable method." If possible, it would be nice to explicitly state the PPV and sensitivity of EH+STRipy given your recommended pathogenic thresholds (and any genotype filters).

We have added the predictive value and sensitivity of the different tools used in supplementary Data 8.

40- "... errors can usually be easily distinguished from true sequence variations if the sequencing depth is sufficient" Would it be possible to recommend a minimum ONT sequencing depth?

Our current read depth, pooling up to 96 samples on a MinION flowcell, ranges from 14,642 to 709,101 (median: 159,437). This depth yields between 86 and 26,380 reads for the larger allele (median: 5,660 reads) and between 3,063 and 192,370 reads for the smaller allele (median: 29,819 reads). As visible on Figure3A-B, the amplification bias between the small and the large alleles depends on the size and motif composition of each allele, making each case unique. However, we estimate that a minimum read depth of 15,000X is required to sequence enough reads for large alleles up to 937 repeats.

41- "Overall, the frequency of pathogenic FGF14 expansions in the control population we studied, which is representative of the population from North Rhine-Westphalia, using 250 repeats as a threshold, is 7 out of 802 (0.87%). In particular, we observed two control individuals (46 and 70 years old) with a pure AAG repeat expansion ≥ 300 (313 and 319 repeats, respectively)." Would it be possible to provide separate penetrance estimates for pure vs. interrupted expansions? I think this could help analysts with interpretation of cases in our rare disease cohorts.

We now show the percentages of pure expansions according to different repeat number ranges in the revised Fig. 1.

Calculating the penetrance estimate for pure versus interrupted alleles is more challenging. The main difficulty lies in determining what truly correspond to interruptions, as the sequence varies between individuals (see response to point 49). Moreover, we do not have access to the neurological status of control individuals, and it is possible that control individuals may exhibit symptoms (such as nystagmus or ataxia) that were not present at their inclusion in the HNR study. Therefore, we think that investigating the penetrance of interrupted alleles necessitates a dedicated study with a larger dataset.

42- "Additional causes include a pathogenic SCA6 expansion." Not immediately clear if this should be read as "SCA6 expansions were found in some of our cases" or "SCA6 is another possible cause of ataxia"

We agree with the reviewer that this sentence was unclear and have removed it. In addition, the discussion has been extensively revised, including a new reference of a study published during the review of this manuscript (Méreaux et al. 2024).

43- ".. size of the smaller allele a .. play an additive effect and partially determine the penetrance of alleles <320 repeats." Out of curiosity, was there any indication that individuals with biallelic expansions (regardless of their AAO) had faster disease progression once symptoms began (which would be consistent with the model from [Kaplan,Shapiro 2007])?

We have now included case reports for all six individuals with biallelic expansions. Among them, three have been identified as clinical outliers, with a more rapid progression, while the other three exhibit a progression similar to individuals with heterozygous repeat expansions. This has been added in the results and discussion. However, we think the data is currently limited for drawing definitive conclusions especially since there are possible confounders accounting for rapid progression (see case reports). But it is indeed possible that some individuals with two large alleles may indeed experience a more rapid progression.

44- Do you have any recommendations for whether the pre-repeat sequence (as determined by short read sequencing) should be looked at during analysis/diagnosis? If yes, how should this be included in decision making?

We have observed a positive association between specific pre-repeat sequences and increased expansion size. However, the significance of these findings remains uncertain. It is possible that certain pre-repeats influence overall stability, or conversely, that increased instability of the repeat promotes the appearance of pre-repeat sequences. Although we do not believe that these sequences should be used for diagnostic purposes at present, we suggest that future studies should consider the sequence of flanking regions and pre-repeats as potential modifiers of phenotype and disease progression (page 16).

Figure 2:

45-What does "VAR/+" mean?

VAR means variant; this has now been added to the legend.

46-What do the "2", "3", "4", "6" labels on some individuals mean?

The numbers in symbols indicate the number of siblings with the same sex; this has now been added to the legend.

47-Caption: Should "Black symbols indicates subjects examined and sampled in the study" be "Black symbols *indicate affected* subjects examined and sampled in the study"?

The sentence has been modified accordingly.

48-Figure 3:

3B: it's hard to distinguish the red color for AGG from the maroon color for ACG. AGG is the same

motif as GAG (yellow) just shifted by 1bp, while ACG is a very different motif. It's currently unclear whether the colors on the red/orange colors on the M95411 and M81457 plots should be interpreted as a mixture of GAA/GAG or GAA/GAC motifs. A farther-from-red color (green?) should clarify this. Also it would be nice if the pathogenic threshold could be placed on these graphs as a vertical line and/or the x-axis labeled as "(bp)" (so it's clear that it's not the # of repeats).

3B: would it make sense to explicitly state in the caption that the pink AAC streaks are likely sequencing errors (as suggested in the Discussion)?

3C: E20-0199 shows an affected individual with a pathogenic-range expansion that's full of interruptions. Is this not meant to be taken as evidence that AAGGAG interruptions are still pathogenic (but have reduced penetrance relative to pure repeats)?

3D: wow, impressive accuracy

3E: It would be helpful to add a vertical line for the pathogenic threshold. So many interesting things in this plot. Can more be said about 1) the single outlier orange dot way above the diagonal? could its read visualizations be shown? Also, it's initially surprising that the orange dots all appear to be clustered together around 1000 bp - is this just that elsewhere they're hidden below the blue dots? The main question this plot raises for me is - why do interrupted repeats appear to show the same (or more) somatic instability compared to pure repeats? I thought a core property of STRs is that interruptions significantly reduce the mutation rate. Is there some way to show that the standard deviation metric is primarily measuring underlying somatic variation and is not a technical artifact of ONT sequencing in long repeats / does not have a high false-positive rate?

We have now changed all plots showing visual representations of nanopore reads included in the manuscript and modified the colors for the ACG motif (now green instead of dark red, see revised Figure 3C).

We have added in the figure legend that AAC streaks are likely errors of sequencing.

We cannot add a vertical line corresponding to the threshold as this would need to take into account the variable flanking regions, which typically vary for each individual. However, we now provide all plots (except the plots with the very high number of repeat, e.g., M100781 and M96642) with the same scale on the X axis for easier comparison.

AAGGAG repeat expansions are actually composed of pure hexameric repeats instead of AAG triplets and do not necessarily correspond to interrupted repeats. These repeats have specific properties different from AAG repeats as shown by their ability to form G4 structures.

49-Figure 4:

4A: can more be said about the interruption patterns referred to as "AAG interruption" and "AAG interruption?" - ideally by providing example read visualizations (or just more detailed descriptions). Currently, it's not clear how to recognize these in our own data, and what the implications are if we do see them. Figure 5 lists single interruptions in detail, but that's further down in the paper, and it's unclear how the "interruption", "interruption?" labels in this figure map to the list in 5A.

4B: the caption uses "n" for the # of individuals, but the plot labels use "n" as allele counts. Might be clearer to put "n = 269 alleles" ..

4D: what is the gray "ns" category in the legend? I don't see it in the plot. Also, would it be possible to have 2 plots like this - one for pure AAG (or all) alleles and one for AAGGAG alleles?

As stated in the results (page 7) and in the methods (page 23), we distinguished "true AAG interruptions (i.e., disrupting repeats in the middle)" from interruptions limited to 3' or 5' sides of the repeats.

Supplementary Figure 5 shows examples of true AAG interruptions (2 panels above), interruptions limited to 3' or 5' sides of the repeats (2 panels in the middle), and a repeat mainly composed of another hexameric (AAGGAG) motif (2 panels below). This is now indicated in the legend.

The exact definition and clinical interpretation of interruptions has to be clarified in further studies including larger cohorts of patients and controls. Our hypothesis is that expansions with interruptions possibly stabilizing repeats may be less penetrant than pure ones. We have added a sentence in the discussion to highlight the necessity to address this question in follow up studies looking for genetic modifiers (page 16).

4B: We modified the legend

4C: We have removed the ns category that was indeed not present in the plot.

50-Figure 5:

5A: what is the "Additional repeat" list? My guess is that these are interspersed within the repeat sequences of some samples, but are rarer than AAGGAG main repeats (and possibly cover less of the repeat sequence?). Would it make sense to combine these with the "single interruption" list into one "scattered interruptions" list?

Additional repeats correspond to motifs different than the main repeat motif but repeated at least twice (e.g., GAGGAG). We considered them different than single motif (e.g., GAG) interruptions. The list of additional repeat sequences appear in Fig. 5A.

The interpretation of what constitutes an "interruption" or not remains unclear at the moment, which is why we decided to separate 'additional repeats' and 'single motifs' interruptions but we agree with the reviewer that their impact may be similar.

For this reason, we chose a more descriptive and systematic approach to describe the variability we observed in repeats. We name 'interruptions' motifs that appear disruptive to the overall structure sequence occurring in the middle while 'interruptions?' corresponds to a category of disruption that is expected to be less disruptive (e.g., motifs located close to flanking regions).

This was now added to the methods and examples of interruptions can be seen in Supplementary Fig. 5.

51-Figure 6:

If possible, I think it would be helpful to show stats for categories where 0 individuals are affected with the given phenotype (for example, the expansion-negative category for Migraine, or the $180 \leq \text{exp} < 250$ for Seizures/epilepsy). Also, out of curiosity, what does it mean that impaired gait, impaired stand, and dysarthria appear to be non-correlated or anti-correlated with expansion size? Finally, in case there are sufficient numbers of individuals for some categories, it would be interesting to see how these phenotypes are associated with interrupted vs. pure expansions.

Based on comments from other reviewers, we have changed the way the clinical data are presented. We have now removed figure 6 to and provide the clinical information as Table 1.

52-Figure 7:

7A: The caption "Comparison of the AAO in patients negative for SCA27A/SCA27B.." seems contradictory with the figure title "progression in patients with SCA27B/SCA27A." I had assumed the "negative" category on the x-axis meant patients that don't have FGF14 expansions or known SCA27A variants.

7F, 7G: a brief summary of what SARA and ICARS scores measure would be appreciated - along with any discussion of why scores appear to be more stable for individuals with > 300 repeats than those with $250 \leq x < 300$

We have changed the title of the figure to better reflect its content.

We have now added the definition of SARA and ICARS scores in the methods and have added the following sentence in the figure legend "SARA and ICARS scores are clinical rating scales that are used for semiquantitative assessment of cerebellar ataxia".

As discussed in the results and discussion, the patients who have a more rapid or severe progression have either biallelic expansions or possible additional diagnoses that may account for it and we do not think the difference relates only to the size of the larger allele.

53-Sup. Figure 1:

if not difficult to add, it would be interesting if colors could be used also in this figure to show which dots represent interrupted vs. pure repeats.

Thank you for this suggestion. The problem here is that we mainly have genome data for patients with ataxia who have pure expansions. Only two individuals have an AAG expansion that may be interrupted at its 5' end and one has an expansion mainly composed of AAGGAG repeats. We think this is too few to draw any conclusions.

54-Sup. Figure 5:

5A: in these box plots, do the dots represent single nanopore reads?

5D: same comment as for figure 4A: I find the "AAG interruption" vs. "AAG interruption?" labels to be confusing. For example, which labels would apply to the interruption patterns shown in patients 998894 and 99855 in Sup. Figure 4?

5A: Dots represent single nanopore read outliers. This has been added in the figure legend.

See response to point 49: Supplementary Figure 4 shows examples of true AAG interruptions (2 panels above), interruptions limited to 3' or 5' sides of the repeats (2 panels in the middle), and a repeat mainly composed of another hexameric (AAGGAG) motif (2 panels below).

55-Sup. Figure 7:

7C: Caption doesn't say what "Fam (IP)" and "Spo" stand for.

"Fam AD" means familial with observed dominant inheritance; "Fam IP" means familial with incomplete inheritance (parents non-affected); "spo" means sporadic cases; this has now been added in the legend of the figure.

56-Case reports:

M83382;#1: sentence with missing phrase: "Genome sequencing including identified a heterozygous splice ..."

The sentence is complete in the revised version.

57-Methods - Targeted sequencing with Oxford Nanopore Technologies (ONT):

"We considered only reads containing 2x AAG (i.e. AAGAAG)." not sure what this means.

We only considered reads with the sequence "AAGAAG" (two AAG repeats) in the analysis. The sentence was modified.

Reviewer #5:

Mohren et al. discuss SCA27B associated AAG/GAA repeat expansion in the intronic region of FGF14 in terms of the repeat number, average onset age, influence of pre-repeat regions and interruptions.

58-Although the authors have put lots of effort in terms of the sample size (both the control and patients), I am not quite convinced about the novelty of the outcome discussed here. As mentioned by the authors themselves in the abstract, the association between the GAA/AAG repeat and SCA27B is already known. This study investigates in detail about the average onset age, influence of pre-repeat regions and interruptions, which is a minimal advancement in understanding the mechanism of SCA27B. Although I don't deny the importance of the results presented here, I don't think this work is suitable for Nature communications.

We hope that the revised version better highlights the novelty of the findings presented in our manuscript. These novel aspects encompass the redefinition of the pathogenic threshold and a comprehensive analysis of the sequence of *FGF14* expansions, elucidating a complexity in sequence previously undescribed. This complexity has implications for delineating pathogenic and non-pathogenic expansions, as well as understanding their penetrance.

See also response to point 5, reviewer 2 for a detailed response.

Minor:

59- the authors should consider mentioning AAG/GAA expansion in Friedreich's ataxia in the manuscript

AAG/GAA expansions causing Friedreich's ataxia are mentioned in the discussion (page 17)

60- line #499: "bioinformatic tool"  how does it differ from STRling?

STRling is one of several bioinformatics tools able to detect repeat expansions.

61- How have the authors verified the quality of the reads?

As indicated in the methods, we use pycoQC (version 2.5.2) and NanoPlot (version 1.41.6) for quality control (page 20). The sequencing depth (median: 159437) also compensates for the possible lower read quality obtained by older versions of Nanopore sequencing.

62- Table 1 and Figure 6  why should there be an AAO for negative?

This study enrolled patients presenting with cerebellar ataxia, who remained negative after short-read exome or genome analysis. We conducted tests for repeat expansions in *FGF14* and compared clinical features and age at onset between patients positive and negative for expansions in *FGF14*. In this context, the negative category refers to patients with cerebellar ataxia who tested negative for expansions in *FGF14* (thus having an age at onset for ataxia).

REVIEWERS' COMMENTS

Reviewer #1 (Remarks to the Author):

The authors answered my comments and questions satisfactorily regarding the previous version. I noticed a verb is missing in the abstract at line 53.

Sincerely,
Wouter De Coster

Reviewer #2 (Remarks to the Author):

The authors have appropriately addressed the majority of my comments, and the revised manuscript has markedly improved. In particular, the clinical data are now much better presented than in the initial version of the manuscript.

The only remaining point I am not entirely convinced of is whether repeats shorter than 250 can be clearly considered pathogenic based on the provided data, which has, as I already mentioned in my initial review, crucial implications for diagnosis and counseling in clinical practice. However, this data is now also much better presented than in the initial version of the manuscript, but this topic still urgently requires further validation in future studies, which the authors should emphasize in their conclusions.

The family data is still not convincing, and investigating the parents of the affected siblings in family SPG76 would have been more meaningful than including their children. However, the authors explained that their conclusions are mainly drawn from the differences between patients and controls, which is now appropriately presented in the results section.

I now agree that the authors provide evidence that repeat expansions below 250 might contribute to clinical manifest ataxia representing something like a risk factor, but other genetic factors might also be required. Accordingly, the authors now report additional genetic factors potentially contributing to the phenotype in the results section in the "Intermediate FGF14 alleles" paragraph. Together, this data is interesting but still somewhat preliminary, and further studies on this topic are required.

Nevertheless, the discussion on this topic is now appropriate. However, in the context of mediating genetic factors, additionally discussing the STUB1/SCA17 story (PMID: 34906452), likewise from the ataxia field, might support the authors' assumptions.

However, contrary to the result part and the discussion, the authors still state, "We suggest that pure AAG expansions are pathogenic from a lower threshold (comprised between 180 and 200 repeats)..." in the final paragraph without mentioning that this repeat size alone might not be directly pathogenic and that this finding requires further investigation. I recommend toning down this statement accordingly by mentioning these limitations.

Reviewer #2 (Remarks on code availability):

NA

Reviewer #3 (Remarks to the Author):

I have reviewed the revised manuscript "NCOMMS-24-08210A" and commend you on addressing the comments thoroughly. Your study provides crucial insights into FGF14 repeat expansions in SCA27B, demonstrating the efficiency of STRling and ExpansionHunter in detecting these expansions from short-read genomic data.

The findings on the lower thresholds for pathogenic AAG expansions and the differentiation between pathogenic and non-pathogenic repeats based on secondary structures are particularly valuable. Your clinical characterization of SCA27B and the recommendation to re-evaluate pathogenic thresholds and integrate expansion sequencing into diagnostics are significant contributions.

Your manuscript presents important and novel information and is a valuable addition to the literature.

Paula Saffie

Reviewer #3 (Remarks on code availability):

Again, I acknowledge the well structured and easy to follow github

Reviewer #4 (Remarks to the Author):

Thank you to the authors for addressing my previous comments. Also, I appreciate the new EH with OT analysis and plots, the analysis of PPV/Specificity, as well as the other modifications to the text and figures. The updated manuscript significantly advances our understanding of how to interpret and diagnose FGF14 expansions, and I recommend publication.

Minor suggestions:

If time allows, it would be great to further disambiguate places in the paper that talk about "SCA27B" as a phenotype vs. a genotype.

The 1st paragraph of the Introduction says that SCA subtypes are named "according to their underlying locus/genetic cause" which matches my working understanding of "SCA27B" as having the definition "autosomal dominant late onset ataxia caused by FGF14 expansion". This is subtly different from the first sentence of the updated Abstract which defines "SCA27B" as just the phenotype "autosomal dominant late-onset cerebellar ataxia (SCA27B)".

This sets up potential subsequent confusion around phrases like "SCA27B-negative" where it's unclear if it means phenotype-negative, expansion-negative, or both.

Similarly, the phrase "FGF14-negative" could reasonably mean "expansion negative" or "expansion and LoF variant negative" (and same issue with "FGF14-positive").

Since a significant focus of the paper is delineating differences across observed genotype and phenotype ranges, I think it's worth adding more verbose labels and descriptions to avoid the potential ambiguities. I list specific places below:

line 249-250: "1) patients with a median number of AAG repeats ≥ 250 (n=42)" - it's not obvious whether "median number" is calculated across one allele or both alleles.

"2) SCA27B-negative patients (n=98; Supplementary Data 4)" - after looking at Supplemental data 4, I think this group is better described as "AAG expansion negative"

so I would propose rewording to something like:

group 1) patients with a long allele ≥ 250 AAG repeats (n=42)

group 2) patients with a long allele < 250 AAG repeats or any number of AAGGAG repeats (n=98)

Also, in Supp. Data 4:

In the "group" column as well as the rest of the paper, I think it would be clearer to replace "FGF14 positive" => "AAG-expansion-positive", and replace "negative" => "AAG-expansion-negative"

In the median_n_repeats column, I interpreted this as the number of repeats of the main_motif. If this is correct, should AAGGAG allele sizes be divided by 2 (ie. 316 x AAGGAG should be recorded as 158)?

The same question applies to the caption of Supp. Figure 2B when describing the red numbers in parentheses.

line 262: "according to the mean repeat number" => "median repeat number" ?

--

line 47: "efficiently" => "effectively" or "accurately" (given that accuracy is more important/interesting than the fact that it's fast/computationally efficient)

line 53: "hexameric expansions equally frequent" => "are equally frequent"

line 264: "undistinguishable" => "indistinguishable"

line 496: "This possibly include" => "This possibly includes"

line 541: "account for 23-31% of patients with adult-onset cerebellar ataxia in European populations " => "account for 23-31% of patients with *unsolved* adult-onset .. " ?

line 543: "while interrupted expansions are less penetrant" -- I'm not sure I understand this point and how/where the paper evaluates penetrance of interrupted expansions.

line 622: "Known repeat expansions were additionally called by ExpansionHunter v5.0 (with and without off-target regions) and STRipy".

Was ExpansionHunter evaluated separately from STRipy? If yes:

- 1) what FGF14 locus definition was used?
- 2) how was the list of off-target regions generated?
- 3) could the variant catalog(s) be shared in the methods or supplement?

line 639: what is meant by "molecular tools" here? I think the caption of Supp Figure 1H has an easier-to-understand description of the approach.

line 672: would put either "subjects" or "samples".. the switch is slightly confusing

Supp. Figure 1B-G. Really appreciate the comprehensive comparisons here. If not difficult, it would be nice to add an x=y diagonal line on these plots, to make it obvious when tools are over/under-estimating the allele size.

Reviewer #1 (Remarks to the Author)

The authors answered my comments and questions satisfactorily regarding the previous version. I noticed a verb is missing in the abstract at line 53.

Sincerely,
Wouter De Coster

Thank you. We have added the missing verb line 53.

Reviewer #2 (Remarks to the Author)

The authors have appropriately addressed the majority of my comments, and the revised manuscript has markedly improved. In particular, the clinical data are now much better presented than in the initial version of the manuscript.

The only remaining point I am not entirely convinced of is whether repeats shorter than 250 can be clearly considered pathogenic based on the provided data, which has, as I already mentioned in my initial review, crucial implications for diagnosis and counseling in clinical practice. However, this data is now also much better presented than in the initial version of the manuscript, but this topic still urgently requires further validation in future studies, which the authors should emphasize in their conclusions.

The family data is still not convincing, and investigating the parents of the affected siblings in family SPG76 would have been more meaningful than including their children. However, the authors explained that their conclusions are mainly drawn from the differences between patients and controls, which is now appropriately presented in the results section.

I now agree that the authors provide evidence that repeat expansions below 250 might contribute to clinical manifest ataxia representing something like a risk factor, but other genetic factors might also be required. Accordingly, the authors now report additional genetic factors potentially contributing to the phenotype in the results section in the "Intermediate FGF14 alleles" paragraph. Together, this data is interesting but still somewhat preliminary, and further studies on this topic are required.

Nevertheless, the discussion on this topic is now appropriate. However, in the context of mediating genetic factors, additionally discussing the *STUB1/SCA17* story (PMID: 34906452), likewise from the ataxia field, might support the authors' assumptions.

However, contrary to the result part and the discussion, the authors still state, "We suggest that pure AAG expansions are pathogenic from a lower threshold (comprised between 180 and 200 repeats)..." in the final paragraph without mentioning that this repeat size alone might not be directly pathogenic and that this finding requires further investigation. I recommend toning down this statement accordingly by mentioning these limitations.

Thank you for these comments.

After discussion with the clinicians who referred family SPG79 to us, we learned that the mother of the two affected siblings presented symptoms suggesting she was also possibly affected. Both parents are unfortunately deceased and genetic testing was therefore not possible. We have modified the pedigree of this family to reflect this new piece of information.

We have modified some sentences in the discussion (page 15) to highlight the points listed above (including now reference for the *STUB1/SCA17* genetic interaction).

Regarding the last paragraph (which summarizes our study), since the sentence already includes the word 'suggest' and we have multiple pieces of evidence in the paper (e.g., segregation in families with values lower than 250; clinical features being similar in patients with more than 250 and less than 250 repeats; and enrichment of expansions in the range of 180-250 in patients) indicating that the threshold of 250 repeats needs to be reconsidered, we decided to remove the threshold values instead of further toning

down the sentence. We believe this topic, including the necessity for additional studies involving a large number of patients and possible genetic modifiers, is now appropriately discussed on page 15.

Reviewer #3 (Remarks to the Author)

I have reviewed the revised manuscript "NCOMMS-24-08210A" and commend you on addressing the comments thoroughly. Your study provides crucial insights into FGF14 repeat expansions in SCA27B, demonstrating the efficiency of STRling and ExpansionHunter in detecting these expansions from short-read genomic data.

The findings on the lower thresholds for pathogenic AAG expansions and the differentiation between pathogenic and non-pathogenic repeats based on secondary structures are particularly valuable. Your clinical characterization of SCA27B and the recommendation to re-evaluate pathogenic thresholds and integrate expansion sequencing into diagnostics are significant contributions.

Your manuscript presents important and novel information and is a valuable addition to the literature.

Paula Saffie

Dear Paula Saffie, thank you for these positive comments.

Reviewer #4 (Remarks to the Author)

Thank you to the authors for addressing my previous comments. Also, I appreciate the new EH with OT analysis and plots, the analysis of PPV/Specificity, as well as the other modifications to the text and figures.

The updated manuscript significantly advances our understanding of how to interpret and diagnose FGF14 expansions, and I recommend publication.

We thank the reviewer once again for their positive evaluation and thorough review of our manuscript. Specific points have been addressed as follows:

Minor suggestions:

If time allows, it would be great to further disambiguate places in the paper that talk about "SCA27B" as a phenotype vs. a genotype.

The 1st paragraph of the Introduction says that SCA subtypes are named "according to their underlying locus/genetic cause" which matches my working understanding of "SCA27B" as having the definition "autosomal dominant late onset ataxia caused by FGF14 expansion". This is subtly different from the first sentence of the updated Abstract which defines "SCA27B" as just the phenotype "autosomal dominant late-onset cerebellar ataxia (SCA27B)".

This sets up potential subsequent confusion around phrases like "SCA27B-negative" where it's unclear if it means phenotype-negative, expansion-negative, or both.

Similarly, the phrase "FGF14-negative" could reasonably mean "expansion negative" or "expansion and LoF variant negative" (and same issue with "FGF14-positive").

Since a significant focus of the paper is delineating differences across observed genotype and phenotype ranges, I think it's worth adding more verbose labels and descriptions to avoid the potential ambiguities. I list specific places below:

line 249-250: "1) patients with a median number of AAG repeats \geq 250 (n=42)" - it's not obvious whether "median number" is calculated across one allele or both alleles.

"2) SCA27B-negative patients (n=98; Supplementary Data 4)" - after looking at Supplemental data 4, I think this group is better described as "AAG expansion negative"

so I would propose rewording to something like:

group 1) patients with a long allele \geq 250 AAG repeats (n=42)

group 2) patients with a long allele $<$ 250 AAG repeats or any number of AAGGAG repeats (n=98)

Also, in Supp. Data 4:

In the "group" column as well as the rest of the paper, I think it would be clearer to replace "FGF14 positive" => "AAG-expansion-positive", and replace "negative" => "AAG-expansion-negative"

We have made changes page 9 to clarify the definition of 'SCA27B-positive' and 'SCA27B-negative' groups according to the reviewer's suggestion. However, we think that these labels actually best define the patients groups, since patients with a pure expansion >250 repeats both have a SCA27B genotype and phenotype; and this is also true for the negative group (both negative).

We would like to avoid using labels such as "expansion-negative" or "AAG-expansion-negative" as the concept of 'expansion', even when considering "pathogenic expansions" is not well defined. Furthermore, we would then need to indicate the gene in addition to the motif i.e., FGF14 AAG expansion-negative as there are other AAG expansions possible in the genome and CTT should actually be the motif used whenever we refer to the gene and not the genomic locus.

'FGF14- positive' and 'FGF14-negative' have been changed to 'SCA27B-positive' and 'SCA27B-negative'.

In the median_n_repeats column, I interpreted this as the number of repeats of the main_motif. If this is correct, should AAGGAG allele sizes be divided by 2 (ie. 316 x AAGGAG should be recorded as 158)? The same question applies to the caption of Supp. Figure 2B when describing the red numbers in parentheses.

We report the number of triplets (rather than the number of repeats) to allow direct comparison of the size of AAG and AAGGAG repeats (i.e. AAGGAG counts as 2 triplets). In Supplementary Data 4, we have now changed the title of column H to 'median_n_triplets' and clarified the numbering of triplets/repeats in the methods.

line 262: "according to the mean repeat number" => "median repeat number" ?

We corrected the sentence.

--

line 47: "efficiently" => "effectively" or "accurately" (given that accuracy is more important/interesting than the fact that it's fast/computationally efficient)

We replaced "efficiently" by "accurately".

line 53: "hexameric expansions equally frequent" => "are equally frequent"

We added the missing verb.

line 264: "undistinguishable" => "indistinguishable"

We corrected this word.

line 496: "This possibly include" => "This possibly includes"

We corrected this typo.

line 541: "account for 23-31% of patients with adult-onset cerebellar ataxia in European populations " => "account for 23-31% of patients with *unsolved* adult-onset .. " ?

"unsolved" was added in the sentence.

line 543: "while interrupted expansions are less penetrant" -- I'm not sure I understand this point and how/where the paper evaluates penetrance of interrupted expansions.

This result come from the patient with biallelic interrupted expansions – we have now made this clear.

line 622: "Known repeat expansions were additionally called by ExpansionHunter v5.0 (with and without off-target regions) and STRipy".

Was ExpansionHunter evaluated separately from STRipy? If yes:

- 1) what FGF14 locus definition was used?
- 2) how was the list of off-target regions generated?
- 3) could the variant catalog(s) be shared in the methods or supplement?

Information of the FGF14 locus definition and off-target catalog was added in the methods (pages 20-21). Variant catalog .json files were uploaded in <https://zenodo.org/records/12542854>.

line 639: what is meant by "molecular tools" here? I think the caption of Supp Figure 1H has an easier-to-understand description of the approach.

We have replaced “molecular tools” by “different methods (nanopore sequencing or fragment analysis)” and rephrased the sentence to enhance clarity.

line 672: would put either "subjects" or "samples" .. the switch is slightly confusing

We now indicate “subjects”.

Supp. Figure 1B-G. Really appreciate the comprehensive comparisons here. If not difficult, it would be nice to add an $x=y$ diagonal line on these plots, to make it obvious when tools are over/under-estimating the allele size.

Plots now include an $x=y$ diagonal line.